# A Mutual Information Perspective on Federated Contrastive Learning

**Christos Louizos, Matthias Reisser, Denis Korzhenkov**
Qualcomm AI Research*
{clouizos,mreisser,dkorzhen}@qti.qualcomm.com

## Abstract

We investigate contrastive learning in the federated setting through the lens of Sim-CLR and multi-view mutual information maximization. In doing so, we uncover a connection between contrastive representation learning and user verification; by adding a user verification loss to each client's local SimCLR loss we recover a lower bound to the global multi-view mutual information. To accommodate for the case of when some labelled data are available at the clients, we extend our SimCLR variant to the federated semi-supervised setting. We see that a supervised SimCLR objective can be obtained with two changes: a) the contrastive loss is computed between datapoints that share the same label and b) we require an additional auxiliary head that predicts the correct labels from either of the two views. Along with the proposed SimCLR extensions, we also study how different sources of non-i.i.d.-ness can impact the performance of federated unsupervised learning through global mutual information maximization; we find that a global objective is beneficial for some sources of non-i.i.d.-ness but can be detrimental for others. We empirically evaluate our proposed extensions in various tasks to validate our claims and furthermore demonstrate that our proposed modifications generalize to other pretraining methods.

## 1 Introduction

For many machine-learning applications "at the edge", data is observed without labels. Consider for example pictures on smartphones, medical data measurements on smart watches or video-feeds from vehicles. Leveraging the information in those data streams traditionally requires labelling - *e.g.* asking users to confirm the identity of contacts in photo libraries, uploading road recordings to a central labelling entity - or the data might remain unused. Fundamentally, labelling data from the edge either happens at the edge or one accepts the communication overhead, privacy costs and infrastructure effort to transfer the data to a central entity and label it there. Labelling at the edge on the other hand either requires enough hardware resources to run a more powerful teacher model or it requires costly end-user engagement with inherent label noise and potential lack of expertise for labelling. Ideally, we can leverage unlabelled data directly at the edge by applying unsupervised learning, without the need for labels nor needing to transfer data to a central location.

In this work, we consider the case of federated unsupervised and semi-supervised learning through the lens of contrastive learning and multi-view mutual information (MI) maximization. The main challenges in this context are twofold: estimating the MI can be difficult because it often requires intractable marginal distributions (Poole et al., 2019). Additionally, the federated environment introduces extra complications, as the global MI objective does not readily decompose into a sum of local (client-wise) loss functions, thereby making it difficult to employ FedAvg (McMahan et al., 2017), the go-to algorithm in federated learning.

To combat these challenges, we introduce specific lower bounds to the global MI that decompose appropriately into local objectives, allowing for straightforward federated optimization. In doing so, we arrive at a principled extension of SimCLR (Chen et al., 2020) to the federated (semi-) unsupervised setting, while uncovering interesting properties. While each user can run vanilla SimCLR locally,

---

*Qualcomm AI Research is an initiative of Qualcomm Technologies, Inc. and/or its subsidiaries.

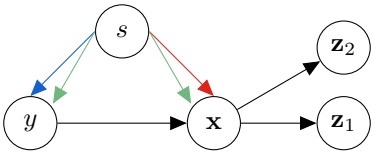

Figure 1: Graphical model of the assumed generative process under the various sources of non-i.i.d.-ness: label-skew, covariate shift and joint shift.

to establish a lower bound for the global MI, it is necessary to add a "user-verification" (UV) loss (Hosseini et al., 2021) for each view. When also dealing with labelled data, the local SimCLR loss on each client needs to contrast datapoints in the batch that belong to the *same* class, thus acting as a form of hard-negative mining. Additionally, besides the UV loss, a label loss is also required for each view. Along with the proposed extensions, we also consider how different sources of non-i.i.d.-ness can impact the performance of federated unsupervised learning through *global* MI maximization. We show that such an objective is beneficial for specific sources of non-i.i.d.-ness but it can be detrimental for others. Finally, while our theoretical analysis and model design was based on SimCLR, we demonstrate that they are generally applicable to other pretraining methods as well, such as spectral contrastive learning (HaoChen et al., 2021) and SimSiam (Chen & He, 2021).

## 2 FEDERATED MULTI-VIEW MUTUAL INFORMATION MAXIMIZATION

Mutual information (MI) has been a paramount tool for unsupervised representation learning; Sim-CLR (Chen et al., 2020), one of the most popular self-supervised learning methods, can be cast as learning an encoder model that maximizes the MI between two views of the same image (Wu et al., 2020). Applying SimCLR to the federated setting however is not straightforward, primarily because the global dataset is not accessible during optimization. In FL, each client only has a subset of the available dataset, and this subset is not necessarily representative of the global dataset due to differences in the data-generative process between clients. Various methods have been proposed to mitigate this effect via global dictionaries of representations (Zhang et al., 2020) or feature alignment regularizers (Wang et al., 2022). In this work, we adopt a different view and extend SimCLR to the federated setting through the lens of global multi-view MI maximization.

### 2.1 FEDERATED SIMCLR

Assume that we have access to an encoder $p_\theta(\mathbf{z}|\mathbf{x})$ with parameters $\theta$. We would like to train this encoder, such that we maximize the MI between the representations of two views of the input $\mathbf{x} \in \mathbb{R}^{D_x}$, namely, $\mathbf{z}_1, \mathbf{z}_2 \in \mathbb{R}^{D_z}$, in the federated setting. Let $s \in \mathbb{N}$ denote the client ID and $p(s)$ a distribution over clients.

In federated learning (FL), the non-i.i.d.-ness can manifest in various ways: a) label skew, where each client $s$ has a different distribution over labels $p(y|s)$ but the same $p(\mathbf{x}|y)$, the most common non-iid-ness assumed in the FL literature, b) covariate shift, where each client has a different distribution over features for a specific class $p(\mathbf{x}|y, s)$, *e.g.* due to different mobile sensors, but the same $p(y)$ and c) joint shift, where both, the distribution of $\mathbf{x}, y$ vary as a function of $s$. This affects the assumed data-generating process of SimCLR representations accordingly, which we illustrate in Figure 1.

Let $\mathrm{I}(x; y)$ denote the MI between $x, y$ and $\mathrm{I}(x; y|z)$ be the MI between $x, y$ conditioned on a third variable $z$. Based on the aforementioned generative process and assuming that all labels are unknown, we start the derivation of federated SimCLR from the chain rule of MI:

$$\mathrm{I}_\theta(\mathbf{z}_1; s, \mathbf{z}_2) = \mathrm{I}_\theta(\mathbf{z}_1; \mathbf{z}_2) + \mathrm{I}_\theta(\mathbf{z}_1; s|\mathbf{z}_2) = \mathrm{I}_\theta(\mathbf{z}_1; s) + \mathrm{I}_\theta(\mathbf{z}_1; \mathbf{z}_2|s) \quad (1)$$

$$\underbrace{\mathrm{I}_\theta(\mathbf{z}_1; \mathbf{z}_2)}_{\text{Global multi-view MI}} = \underbrace{\mathrm{I}_\theta(\mathbf{z}_1; \mathbf{z}_2|s)}_{\text{Local multi-view MI}} + \underbrace{\mathrm{I}_\theta(\mathbf{z}_1; s)}_{\text{Client ID MI}} - \underbrace{\mathrm{I}_\theta(\mathbf{z}_1; s|\mathbf{z}_2)}_{\text{Excess client ID MI}}. \quad (2)$$

We see that the multi-view MI in the federated setting decomposes into three terms; we want to maximize the average, over the clients, local MI between the representations of the two views $\mathbf{z}_1, \mathbf{z}_2$, along with the MI between the representation $\mathbf{z}_1$ and the client ID $s$ while simultaneously minimizing the additional information $\mathbf{z}_1$ carries about $s$ conditioned on $\mathbf{z}_2$. Such MI decompositions have

also been considered at Sordoni et al. (2021) for improving MI estimation in a different context. Unfortunately, in our case these terms require access to potentially intractable or hard to obtain distributions, so we will resort to easy to compute and evaluate variational bounds.

For the first term, *i.e.*, the client conditional MI between the two views, we provide proposition 1 which uses the standard InfoNCE bound (Poole et al., 2019), leading to an objective that decomposes into a sum of local terms, one for each client, thus allowing for federated optimization with FedAvg.

**Proposition 1.** *Let $s \in \mathbb{N}$ denote the user ID, $\mathbf{x} \in \mathbb{R}^{D_x}$ the input and $\mathbf{z}_1, \mathbf{z}_2 \in \mathbb{R}^{D_z}$ the latent representations of the two views of $\mathbf{x}$ given by the encoder with parameters $\theta$. Given a critic function $f : \mathbb{R}^{D_z} \times \mathbb{R}^{D_z} \to \mathbb{R}$, we have that*

$$\mathrm{I}_\theta(\mathbf{z}_1; \mathbf{z}_2 | s) \geq \mathbb{E}_{p(s)p_\theta(\mathbf{z}_1, \mathbf{z}_2 | s)_{1:K}} \left[ \frac{1}{K} \sum_{k=1}^{K} \log \frac{\exp(f(\mathbf{z}_{1k}, \mathbf{z}_{2k}))}{\frac{1}{K} \sum_{j=1}^{K} \exp(f(\mathbf{z}_{1j}, \mathbf{z}_{2k}))} \right]. \tag{3}$$

All of the proofs can be found in the appendix. This corresponds to a straightforward application of SimCLR to the federated setting where each client performs SimCLR training locally, *i.e.*, clients contrast against their local dataset instead of the global dataset. We will refer to this objective as *Local SimCLR*.

In order to optimize the global MI instead of the local MI, we need to address the two remaining terms of equation 2. The first term, $\mathrm{I}_\theta(\mathbf{z}_1; s)$, requires information from the entire federation, *i.e.*, $p_\theta(\mathbf{z}_1)$, which is intractable. However, with lemma 2.1 we show that by introducing a "client classification" task, we can form a simple and tractable lower bound to this term.

**Lemma 2.1.** *Let $s \in \mathbb{N}$ denote the client ID, $\mathbf{x} \in \mathbb{R}^{D_x}$ the input and $\mathbf{z}_1 \in \mathbb{R}^{D_z}$ the latent representation of a view of $\mathbf{x}$ given by the encoder with parameters $\theta$. Let $\phi$ denote the parameters of a client classifier $r_\phi(s | \mathbf{z}_1)$ that predicts the client ID from this specific representation and let $\mathrm{H}(s)$ be the entropy of the client distribution $p(s)$. We have that*

$$\mathrm{I}_\theta(\mathbf{z}_1; s) \geq \mathbb{E}_{p(s)p_\theta(\mathbf{z}_1 | s)} \left[ \log r_\phi(s | \mathbf{z}_1) \right] + \mathrm{H}(s) \tag{4}$$

With this bound we avoid the need for the intractable marginal $p_\theta(\mathbf{z}_1)$ and highlight an interesting connection between self-supervised learning in FL and user-verification models (Yu et al., 2020; Hosseini et al., 2021). For the last term of equation 2, we need an upper bound to maintain an overall lower bound to $\mathrm{I}_\theta(\mathbf{z}_1; \mathbf{z}_2)$. Upper bounds to the MI can be problematic as they require explicit densities (Poole et al., 2019). Fortunately, in our specific case, we show in lemma 2.2 that with an additional client classification task for the second view, we obtain a simple and tractable upper bound.

**Lemma 2.2.** *Let $s \in \mathbb{N}$ denote the user ID, $\mathbf{x} \in \mathbb{R}^{D_x}$ the input and $\mathbf{z}_1, \mathbf{z}_2 \in \mathbb{R}^{D_z}$ the latent representations of the views of $\mathbf{x}$ given by the encoder with parameters $\theta$. Let $\phi$ denote the parameters of a client classifier $r_\phi(s | \mathbf{z}_2)$ that predicts the client ID from the representations. We have that*

$$\mathrm{I}_\theta(\mathbf{z}_1; s | \mathbf{z}_2) \leq -\mathbb{E}_{p(s)p_\theta(\mathbf{z}_2 | s)} \left[ \log r_\phi(s | \mathbf{z}_2) \right] \tag{5}$$

By combining our results, we arrive at the following lower bound for the global MI that decomposes into a sum of local objectives involving the parameters $\theta, \phi$. We dub it as *Federated SimCLR*.

$$\mathrm{I}_\theta(\mathbf{z}_1; \mathbf{z}_2) \geq \mathbb{E}_{p(s)p_\theta(\mathbf{z}_1, \mathbf{z}_2 | s)_{1:K}} \left[ \frac{1}{K} \sum_{k=1}^{K} \log \frac{\exp(f(\mathbf{z}_{1k}, \mathbf{z}_{2k}))}{\frac{1}{K} \sum_{j=1}^{K} \exp(f(\mathbf{z}_{1j}, \mathbf{z}_{2k}))} \right.$$
$$\left. + \log r_\phi(s | \mathbf{z}_{1k}) + \log r_\phi(s | \mathbf{z}_{2k}) \right] + \mathrm{H}(s). \tag{6}$$

In this way, Federated SimCLR allows for a straightforward optimization of $\theta, \phi$ with standard FL optimization methods, such as Reddi et al. (2020), and inherits their convergence guarantees. Furthermore, it is intuitive; each client performs locally SimCLR, while simultaneously training a shared classifier that predicts their user ID from both views. The additional computational overhead of this classifier is relatively minor compared to the encoder itself, making it appropriate for resource constrained devices.

**Optimizing the user-verification loss**   For the client ID loss we use a single linear layer followed by softmax with three important modifications, as the *local* optimization of the client ID loss is prone to bad optima due to having "labels" from only "a single class" (that of the client optimizing it) (Yu et al., 2020); a) the linear layer does not have a bias, as that would make the local optimization of the UV loss trivial and would not meaningfully affect the encoder, b) both the inputs to the linear layer as well as the linear layer weights are constrained to have unit norm and, c) each client locally optimizes only their associated vector weight in the linear classifier while all of the others are kept fixed. In this way each client needs to find their "own cluster center" to optimize the UV loss locally. These centers need to be sufficiently far from the cluster centers of the other clients that a client receives from the server and keeps fixed throughout local optimization.

**Effects of non-i.i.d.-ness on the performance on downstream tasks**   Given access to both the global and local MI objectives, we now want to understand how the type of non-i.i.d.-ness determines whether a specific objective is the better choice. To answer this question, we first show at proposition 2 that in the case of label skew, the client classification objective is a lower bound to the MI between the representations $\mathbf{z}_1, \mathbf{z}_2$ and the unavailable label $y$.

**Proposition 2.** *Consider the label skew data-generating process for federated SimCLR from Figure 1 with $s \in \mathbb{N}$ denoting the user ID with $\mathrm{H}(s)$ being the entropy of $p(s)$, $\mathbf{x} \in \mathbb{R}^{D_x}$ the input, $\mathbf{z}_1, \mathbf{z}_2 \in \mathbb{R}^{D_z}$ the latent representations of the two views of $\mathbf{x}$ given by the encoder with parameters $\theta$. Let $y$ be the label and let $r_\phi(s|\mathbf{z}_i)$ be a model with parameters $\phi$ that predicts the user ID from the latent representation $\mathbf{z}_i$. In this case, we have that*

$$\mathrm{I}_\theta(\mathbf{z}_1; y) + \mathrm{I}_\theta(\mathbf{z}_2; y) \geq \mathbb{E}_{p(s)p_\theta(\mathbf{z}_1, \mathbf{z}_2|s)} \left[ \log r_\phi(s|\mathbf{z}_1) + \log r_\phi(s|\mathbf{z}_2) \right] + 2\mathrm{H}(s). \tag{7}$$

Therefore, when the source of non-i.i.d.-ness is heavily dependent on the actual downstream task, the additional client classification objective stemming from the global MI bound is beneficial as it is a good proxy for the thing we care about. In the case of covariate shift, we know that the source of non-i.i.d.-ness is independent of the label, *i.e.*, $\mathrm{I}(y; s) = 0$, so the additional client classification term can actually become detrimental; the representation will encode information irrelevant for the downstream task and, depending on the capacity of the network and underlying trade-offs, can lead to worse task performance. In this case, optimizing the local MI is expected to work better, as the client specific information (*i.e.*, the irrelevant information) is not encouraged in the representations.

## 2.2   Federated Semi-Supervised SimCLR

In practice, labeled data for a specific task are sometimes available. These could for example constitute a curated dataset at the server or a small labelled subset of data on each client. In this case, it will generally be beneficial for the downstream task if the objective takes these labels into account. To this end, we can use the following label-dependent expression for the client conditional MI

$$\mathrm{I}_\theta(\mathbf{z}_1; \mathbf{z}_2|s) = \mathrm{I}_\theta(\mathbf{z}_1; y|s) + \mathrm{I}_\theta(\mathbf{z}_1, \mathbf{z}_2|y, s) - \mathrm{I}_\theta(\mathbf{z}_1; y|s, \mathbf{z}_2). \tag{8}$$

Therefore, once we obtain a label-specific lower bound for this quantity, it will be straightforward to translate it to a label-specific lower bound for the global MI by adding back the user-verification losses for the two views. For the following we will assume that we have an underlying classification task, hence a label $y \in \mathbb{N}$.

For the MI between the two views $\mathbf{z}_1, \mathbf{z}_2$ conditioned on the label $y$ and client $s$, we can make use of proposition 1 by treating $s, y$ as the conditioning set. In this case, we again use the InfoNCE loss, with the exception that we now contrast between datapoints that also belong to the same class,

$$\mathrm{I}_\theta(\mathbf{z}_1; \mathbf{z}_2|y, s) \geq \mathbb{E}_{p(s,y)p_\theta(\mathbf{z}_1, \mathbf{z}_2|y,s)_{1:K}} \left[ \frac{1}{K} \sum_{k=1}^{K} \log \frac{\exp(f(\mathbf{z}_{1k}, \mathbf{z}_{2k}))}{\frac{1}{K} \sum_{j=1}^{K} \exp(f(\mathbf{z}_{1j}, \mathbf{z}_{2k}))} \right]. \tag{9}$$

For the other two terms that involve the label $y$ we can proceed in a similar manner to the client ID $s$. For the MI between $\mathbf{z}_1$ and $y$ conditioned on $s$, as $y$ is also discrete, we can make use of lemma 2.1 by treating $y$ as $s$. Therefore, we introduce a classifier $r_\phi(y|\mathbf{z}_1)$ and obtain the following lower bound

$$\mathrm{I}_\theta(\mathbf{z}_1; y|s) \geq \mathbb{E}_{p(s)p_\theta(y,\mathbf{z}_1|s)} \left[ \log r_\phi(y|\mathbf{z}_1) \right] + \mathrm{H}(y|s), \tag{10}$$

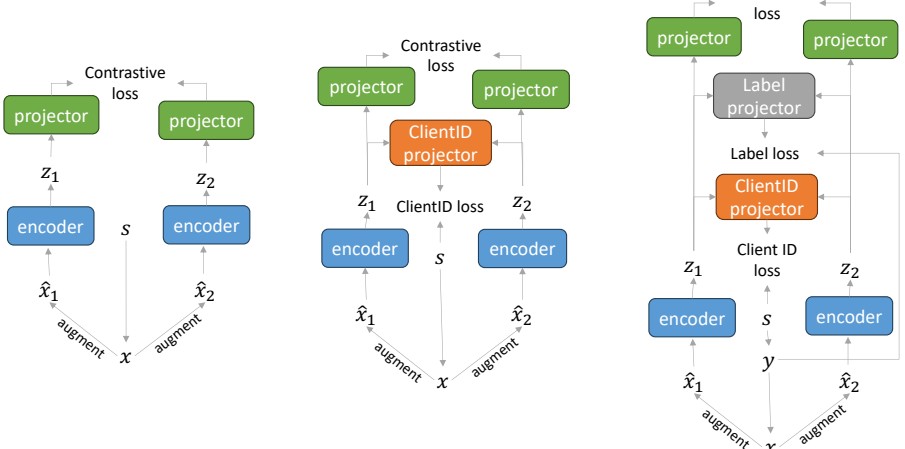

Figure 2: Overview of the SimCLR architectures considered. **Local SimCLR (left)**: each client optimizes a contrastive loss on their own data, thus the federation implicitly optimizes a lower bound to $\mathrm{I}(\mathbf{z}_1; \mathbf{z}_2|s)$. **Federated SimCLR (center)**: along with the contrastive loss on their own data, each client also optimizes a client classifier, thus the federation implicitly optimizes a lower bound to $\mathrm{I}(\mathbf{z}_1; \mathbf{z}_2)$. **Supervised federated SimCLR (right)**: a label-dependent variant of federated SimCLR that encourages clustering according to the label while also optimizing a lower bound to $\mathrm{I}(\mathbf{z}_1; \mathbf{z}_2)$.

where $\mathrm{H}(y|s)$ denotes the entropy of the label marginal at the client, $p(y|s)$. For the MI between $\mathbf{z}_1$ and $y$ conditioned on $\mathbf{z}_2$ and $s$ we make use of lemma 2.2 and get the following upper bound

$$\mathrm{I}_\theta(\mathbf{z}_1; y|\mathbf{z}_2, s) \leq -\mathbb{E}_{p(s,y)p_\theta(\mathbf{z}_2|y,s)} \left[\log r_\phi(y|\mathbf{z}_2)\right]. \tag{11}$$

Putting everything together, we arrive at the following label-dependent lower bound for local SimCLR

$$\mathrm{I}_\theta(\mathbf{z}_1; \mathbf{z}_2|s) \geq \mathbb{E}_{p(s,y)p_\theta(\mathbf{z}_1,\mathbf{z}_2|y,s)_{1:K}} \left[ \frac{1}{K} \sum_{k=1}^{K} \log \frac{\exp(f(\mathbf{z}_{1k}, \mathbf{z}_{2k}))}{\frac{1}{K}\sum_{j=1}^{K} \exp(f(\mathbf{z}_{1j}, \mathbf{z}_{2k}))} \right.$$
$$\left. + \log r_\phi(y|\mathbf{z}_{1k}) + \log r_\phi(y|\mathbf{z}_{2k}) + \mathrm{H}(y|s) \right], \tag{12}$$

which decomposes into intuitive terms; we are performing InfoNCE between the views of the datapoints that belong to the same class and client, while simultaneously trying to predict the class from the representations of both views. To transition from a label-dependent bound for the local SimCLR to a label-dependent bound of the federated SimCLR, it suffices to add the client classifiers

$$\mathrm{I}_\theta(\mathbf{z}_1; \mathbf{z}_2) \geq \mathbb{E}_{p(s,y)p_\theta(\mathbf{z}_1,\mathbf{z}_2|y,s)_{1:K}} \left[ \frac{1}{K} \sum_{k=1}^{K} \log \frac{\exp(f(\mathbf{z}_{1k}, \mathbf{z}_{2k}))}{\frac{1}{K}\sum_{j=1}^{K} \exp(f(\mathbf{z}_{1j}, \mathbf{z}_{2k}))} + \log r_\phi(s|\mathbf{z}_{1k}) \right.$$
$$\left. + \log r_\phi(s|\mathbf{z}_{2k}) + \log r_\phi(y|\mathbf{z}_{1k}) + \log r_\phi(y|\mathbf{z}_{2k}) + \mathrm{H}(y|s) \right] + \mathrm{H}(s). \tag{13}$$

Figure 2 visualizes all of the SimCLR architectures considered in this work.

**The case of unlabelled data** The primary motivation of the previous discussion is to tackle the semi-supervised case, *i.e.*, the case when some clients do not have access to all labels. A simple way to handle the unlabelled data is to fall back to the bound of proposition 1 for the conditional MI when we do not have access to labels. In this way, each client can do a form of "more difficult" contrastive learning for their labelled data, where they contrast against datapoints which are more semantically similar (*i.e.*, they share the same class), while simultaneously trying to predict the correct class whereas for their unlabelled data, they perform standard contrastive learning.

**Label-dependent vs label-independent bound**   Even though both our label-dependent and label-independent bounds are lower bounds of the MI between the representations of the two views, the former should be preferred if labels are available. This is because the label independent one can be satisfied without necessarily clustering the representations semantically, whereas the label dependent one directly encourages clustering according to the label through the additional classification losses, so it is expected to perform better for downstream tasks.

# 3   RELATED WORK

Unsupervised learning in the federated context has gained significant attention in recent years. On the contrastive learning side, Zhang et al. (2020) introduces FedCA, a SimCLR variant for federated setting. The main idea is that the representations between the clients can become misaligned due to the non-i.i.d. nature of FL. The authors then introduce a global dictionary of representations which is shared between all participants and is used to align the representation spaces. One of the main drawbacks of this method is that it requires the transmission of data representations of clients, which leads to reduced privacy. Compared to a global dictionary module, our federated SimCLR aligns the representations of the clients through the additional UV loss component, requiring the communication of just some additional model parameters and not raw representations. Dong & Voiculescu (2021) introduces FedMoCo, an extension of MoCo (He et al., 2020) to the federated setting. Similar to FedCA, FedMoCo shares additional client metadata, *i.e.*, moments of the local feature distributions, from the clients to the server, thus leading to reduced privacy. Li et al. (2023a) also extends MoCo to the federated setting however, instead of using a FedAvg type of protocol, the authors employ a split learning (Poirot et al., 2019) protocol, which leads to reduced compute requirements at the edge but also requires communicating raw representations of the local data to the server. Finally, the closest to our work is the work of Wang et al. (2022) where the authors also explore the effects of non-i.i.d.-ness when training a model with SimCLR in the federated setting. The authors further propose an extension that uses multiple models and encourages feature alignment with an additional loss function. In contrast to FeatARC where the feature alignment loss is added ad-hoc to SimCLR, we can see that from our MI perspective on SimCLR, a feature alignment loss naturally manifests via an additional user-verification loss to SimCLR when optimizing a lower bound to the global MI.

On the non-contrastive learning side, Makhija et al. (2022) introduces Hetero-SSFL, an extension of BYOL (Grill et al., 2020) and SimSiam (Chen & He, 2021) to the federated setting where each client can have their own encoder model but, in order to align the local models, an additional public dataset is required. Zhuang et al. (2022) introduces FedEMA, where a hyperparameter of BYOL is adapted in a way that takes into account the divergence of the local and global models. In contrast to these methods which require several tricks for improved performance, *i.e.*, moving average updates, custom type of aggregations and stop gradient operations, our federated SimCLR method works by just optimizing a straightforward loss function with the defacto standard, FedAvg. On a different note, Lu et al. (2022) proposes to train a model with pseudo-labels for the unlabelled data and then recover the model for the desired labels via a post-processing step. Finally Lubana et al. (2022) proposes an unsupervised learning framework through simultaneous local and global clustering, which requires communicating client data representations, *i.e.*, the cluster centroids, to the server.

On the federated semi-supervised learning side, most works rely on generating pseudo-labels for the unlabelled examples. Jeong et al. (2020) proposes FedMatch, an adaptation of FixMatch (Sohn et al., 2020) to the federated setting by adding one more consistency loss that encourages the models learned on each client to output similar predictions for the local data. The authors also propose a pseudo-labelling strategy that takes into account the agreement of client models and a parameter decomposition strategy that allocates separate parameters to be optimized on unlabelled and labelled data. In contrast, our semi-supervised objectives are simpler, do not rely on pseudo-labels (which introduce additional hyper-parameters for filtering low-confidence predictions) and do not require communicating client specific models among the federation. Liang et al. (2022) proposes a student-teacher type scheme for training on unlabelled data, where consistency regularization is applied. The teacher model is an exponential moving average of the student and a novel aggregation mechanism is introduced. Our proposed methods for semi-supervised learning could potentially also benefit from better aggregation mechanisms, but we leave such an exploration for future work. Finally, Kim et al. (2022) introduces ProtoFSSL, which incorporates knowledge from other clients in the local training via sharing prototypes between the clients. While such prototypes do improve performance, they also

reveal more information about the local data of each client, thus reducing privacy. In contrast, our federated semi-supervised framework does not rely on sharing prototypes between the clients.

# 4 EXPERIMENTS

Our experimental evaluation consist of unsupervised and semi-supervised experiments, where for the latter each client has labels for $10\%$ of their data. To quantify the quality of the learned representations, we adapt the classical evaluation pipeline of training a linear probe (LP) to be in line with common assumptions of self-supervised learning. In the unsupervised case, we report the LP accuracy on the union of clients' labelled version of their data, as this corresponds to the traditional non-federated evaluation pipeline. For the semi-supervised case, we train a LP on top of the representations of the clients' labelled training data (which is a subset of the full training set) and then report its test accuracy. At every evaluation for plotting of learning curves, we initialize the LP from the final parameters of the previous evaluation. Furthermore, as we mention at section 2.1, the nature of non-i.i.d. data in FL can manifest in various ways: label skew, covariate shift and joint shift, *i.e.*, a combination of the two. We therefore evaluate, besides label skew (the predominant type of non-i.i.d.-ness assumed in the FL literature), covariate shift by creating a rotated version of CIFAR10 and CIFAR100 as well as a joint shift case where both sources of non-i.i.d.-ness are present. For CIFAR 10 we consider 100 clients whereas for CIFAR100 we consider 500 clients. For the encoder we use a ResNet18 architecture adapted for the CIFAR datasets where, following Hsieh et al. (2020), we replace batch normalization (Ioffe & Szegedy, 2015) with group normalization (Wu & He, 2018).

In order to demonstrate the general usefulness of our theoretical results and model design stemming from our MI perspective, we include two more methods in our evaluation besides SimCLR. The first one is spectral contrastive learning (HaoChen et al., 2021) (dubbed as Spectral CL) as another instance of constrastive learning and the other is SimSiam (Chen & He, 2021), a non-contrastive method. For both of these methods, we consider both a "local" variant where each of the losses is optimized locally and Reddi et al. (2020) is applied to the parameters as well as, based on the intuition from our federated SimCLR, a "global" variant where the same UV loss component of federated SimCLR is added to the baselines. As we show in proposition 2, such an auxiliary task is beneficial in the case of label skew in general. Furthermore we also extend these baselines to the semi-supervised setting. Based on the insights from our label-dependent MI bounds for SimCLR, we consider label-dependent variants of SimSiam and Spectral CL where, when labels are available, the unsupervised losses are evaluated between elements that share the same class and a classification loss for the two views is added to the overall loss function.

**Unsupervised setting** The results in the unsupervised setting can be seen in Table 1. In the case of label skew, adding our user-verification loss to each of the local losses leads to (sometimes dramatic) improvements in all cases. This is to be expected, as in this case the mutual information between the labels and the client ID, $I(y; s)$, is quite high, so the UV loss acts as a good proxy for the downstream task. For SimCLR we observe a $\sim 6\%$ improvement on CIFAR 10/100 and on Spectral CL we observe $\sim 11\%$ and $\sim 8\%$ respectively. SimSiam type of methods generally underperformed compared to SimCLR and Spectral CL, and we believe this is due to representation collapse, especially given that in our setting we employ group normalization instead of batch-normalization. On covariate shift, we now see that the situation is flipped; as in this case $I(y; s) = 0$, local SimCLR / Spectral CL are doing better compared to their global counterparts that include the UV loss. Both local SimCLR and Spectral CL perform better by $\sim 1-2\%$ and $\sim 2-4\%$ on CIFAR 10 and CIFAR 100 respectively, with local SimCLR providing the better overall performance. Finally, on the joint shift case, the label skew is strong enough to allow for improvements with the additional UV loss components in most cases; for SimCLR there is an improvement of $\sim 4-5\%$ and for Spectral CL there is a $\sim 8\%$ improvement for CIFAR 10 but a drop of $\sim 8\%$ for CIFAR 100. We attribute the latter to the overall instability of Spectral CL in our CIFAR 100 experiments, explained by the large standard error.

Overall, we observe that the results are consistent with our expectations; when the source of non-i.i.d.-ness in the federated setting is strongly correlated with the downstream task, optimizing a "global" objective, such as $I(\mathbf{z}_1, \mathbf{z}_2)$, is beneficial, as the additional UV term serves for a good proxy for the downstream task. This intuition also generalizes to one of our baselines as, *e.g.*, even Spectral CL benefits from the addition of the UV loss in such settings. In the absence of such correlation, the

Table 1: Test set performance (%) on the unsupervised setting along with standard error over 5 seeds. Clients' data is assumed to be fully annotated for LP fine-tuning in the unsupervised case.

| | CIFAR 10 | | | CIFAR 100 | | |
|---|---|---|---|---|---|---|
| Method | Label skew | Covariate shift | Joint shift | Label skew | Covariate shift | Joint shift |
| Local SimCLR | $79.4_{\pm 0.2}$ | $\mathbf{74.3_{\pm 0.3}}$ | $71.0_{\pm 0.4}$ | $42.2_{\pm 0.2}$ | $\mathbf{41.2_{\pm 0.2}}$ | $38.1_{\pm 0.3}$ |
| Federated SimCLR | $\mathbf{85.0_{\pm 0.2}}$ | $73.8_{\pm 0.2}$ | $\mathbf{74.8_{\pm 0.5}}$ | $\mathbf{48.5_{\pm 0.1}}$ | $39.5_{\pm 0.2}$ | $\mathbf{43.1_{\pm 0.2}}$ |
| Spectral CL | $76.5_{\pm 1.1}$ | $\mathbf{73.5_{\pm 0.4}}$ | $68.2_{\pm 0.6}$ | $33.3_{\pm 6.0}$ | $\mathbf{33.6_{\pm 2.3}}$ | $\mathbf{29.6_{\pm 6.2}}$ |
| Spectral CL + UV | $\mathbf{87.8_{\pm 0.3}}$ | $71.7_{\pm 0.5}$ | $\mathbf{76.6_{\pm 0.6}}$ | $\mathbf{41.0_{\pm 6.4}}$ | $29.3_{\pm 4.8}$ | $21.5_{\pm 6.2}$ |
| SimSiam | $\mathbf{40.0_{\pm 0.5}}$ | $\mathbf{39.9_{\pm 0.3}}$ | $\mathbf{39.6_{\pm 0.3}}$ | $\mathbf{16.9_{\pm 0.3}}$ | $\mathbf{16.6_{\pm 0.4}}$ | $\mathbf{16.9_{\pm 0.4}}$ |
| SimSiam + UV | $35.4_{\pm 0.4}$ | $35.4_{\pm 0.2}$ | $34.5_{\pm 0.3}$ | $16.5_{\pm 0.2}$ | $16.5_{\pm 0.3}$ | $16.3_{\pm 0.5}$ |
| Supervised | $89.6_{\pm 0.1}$ | $78.3_{\pm 0.4}$ | $76.3_{\pm 1.1}$ | $59.2_{\pm 0.2}$ | $47.9_{\pm 0.2}$ | $43.9_{\pm 0.3}$ |

Table 2: Test set performance (%) on the semi-supervised setting with 10% labelled data on each client along with standard error over 5 seeds. We use the corresponding labelled subset for the LP.

| | CIFAR 10 | | | CIFAR 100 | | |
|---|---|---|---|---|---|---|
| Method | Label skew | Covariate shift | Joint shift | Label Skew | Covariate shift | Joint shift |
| Local SimCLR | $74.5_{\pm 0.3}$ | $\mathbf{49.1_{\pm 1.3}}$ | $45.8_{\pm 1.4}$ | $30.3_{\pm 0.2}$ | $\mathbf{15.1_{\pm 0.4}}$ | $13.1_{\pm 0.3}$ |
| Federated SimCLR | $\mathbf{78.0_{\pm 0.2}}$ | $\mathbf{50.3_{\pm 1.1}}$ | $\mathbf{49.9_{\pm 1.4}}$ | $\mathbf{34.5_{\pm 0.3}}$ | $14.8_{\pm 0.3}$ | $\mathbf{14.6_{\pm 0.3}}$ |
| Spectral CL | $74.2_{\pm 0.3}$ | $48.0_{\pm 0.7}$ | $45.4_{\pm 1.5}$ | $30.1_{\pm 0.2}$ | $\mathbf{14.1_{\pm 0.4}}$ | $12.3_{\pm 0.3}$ |
| Spectral CL + UV | $\mathbf{79.6_{\pm 0.3}}$ | $\mathbf{49.7_{\pm 1.0}}$ | $\mathbf{49.8_{\pm 1.1}}$ | $\mathbf{34.0_{\pm 0.2}}$ | $13.7_{\pm 0.3}$ | $\mathbf{13.6_{\pm 0.4}}$ |
| SimSiam | $75.3_{\pm 0.4}$ | $46.8_{\pm 0.7}$ | $40.5_{\pm 0.9}$ | $30.7_{\pm 0.2}$ | $13.4_{\pm 0.3}$ | $12.8_{\pm 0.3}$ |
| SimSiam + UV | $\mathbf{80.4_{\pm 0.2}}$ | $\mathbf{50.0_{\pm 1.2}}$ | $\mathbf{44.3_{\pm 1.0}}$ | $\mathbf{34.3_{\pm 0.1}}$ | $\mathbf{13.6_{\pm 0.3}}$ | $\mathbf{14.0_{\pm 0.4}}$ |
| Supervised | $75.1_{\pm 0.2}$ | $48.1_{\pm 0.9}$ | $42.7_{\pm 1.7}$ | $29.6_{\pm 0.3}$ | $12.6_{\pm 0.2}$ | $12.2_{\pm 0.1}$ |

simple local SimCLR / Spectral CL variants are doing better since they do not encode information in the representations that is irrelevant for the downstream task.

**Semi-supervised setting** Our semi-supervised results with 10% labelled data in Table 2 show interesting observations. Overall, we improve performance with semi-supervised training relative to purely supervised training on the labelled subset of the data. On CIFAR 10, we notice that our semi-supervised models with the UV loss do better than the local variants on all sources of non-i.i.d.-ness, even in the case of covariate shift. Despite the limited quantity of labels available, we believe that the encoders possessed sufficient capacity to both retain and separate the label-specific and label-independent (*e.g.*, rotation) information. Consequently, the downstream LP could accurately use the label-specific portion of the representations for its predictions. SimSiam does much better in this setting, as the supervised objective prevented representation collapse, achieving the best performance on label skew when we add the UV loss, whereas Federated SimCLR does best on the joint shift.

### 4.1 ABLATION STUDIES

In this section we perform additional experiments in order to investigate the behaviour of local and federated SimCLR under different settings. We adopt our CIFAR 10 setting with 100 clients and strong ($\alpha = 0.1$) joint shift, unless mentioned otherwise.

**Amount of non-i.i.d.-ness** For the first set of experiments we investigate how the amount of non-i.i.d.-ness affects the local and federated SimCLR performance with $E = 1$. We adopt the joint shift setting and perform experiments with different strengths for each source of non-i.i.d.-ness. The results can be seen in Figure 3a where we have an interesting observation; federated SimCLR does *better* the *higher* the amount of label skew non-i.i.d.-ness is, in fact even surpassing the performance of local SimCLR on i.i.d. data. This can be explained from our proposition 2. As the amount of label skew increases, the client ID carries more information about $y$, thus $I_\theta(\mathbf{z}_1, y|s)$ becomes lower and the lower bound tighter. On the flipside, when there is strong covariate shift and not enough label-skew, we observe that local SimCLR has consistently better performance.

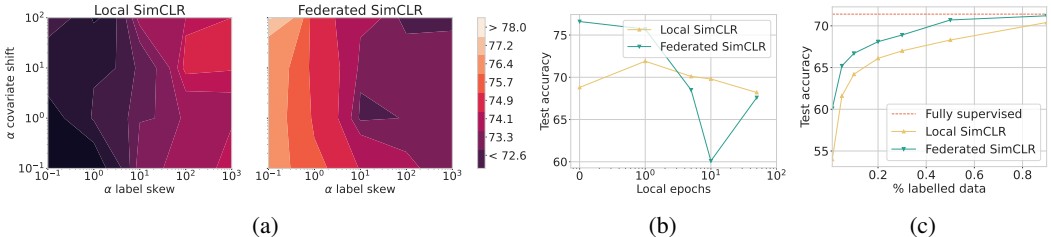

Figure 3: CIFAR 10 ablation studies. (a) Performance of local and federated SimCLR as a function of the non-i.i.d.-ness strength $\alpha$ for covariate shift and label skew. (b) Performance of local and federated SimCLR for different amount of local epochs $E$ in the case of strong ($\alpha = 0.1$) covariate shift and label skew. (c) Performance of local and federated SimCLR in the semi-supervised setting as a function of the amount of available labelled data.

**Amount of local updates**    The auxiliary UV objective in federated SimCLR can be problematic for a large amount of local updates, as there is only a single available class at each client. Therefore, federated SimCLR requires relatively frequent synchronization. We show in Figure 3b how the amount of local epochs affect local and federated SimCLR when keeping a fixed computation budget; more local epochs imply less communication rounds and vice versa. We can see that federated SimCLR achieves the best performance of the two with 1 local step, however, its performance drops with more local updates and eventually becomes worse or comparable to local SimCLR.

**Amount of labelled data for the semi-supervised setting**    Finally, we also measure the impact of the amount of available labelled data in the semi-supervised setting for local and federated SimCLR. We measure this by keeping a fixed and labelled holdout set which we use to train a LP on top of the representations given by the two algorithms. We also train a fully supervised (*i.e.*, on $100\%$ labelled training data) baseline with the same augmentations as the SimCLR variants. We can see in Figure 3c that the test accuracy of the LP improves with more labelled data for both algorithms, as expected. Federated SimCLR demonstrates improved performance compared to local SimCLR on all cases considered, with the biggest advantages seen when the amount of available labelled data during training is low. Furthermore, federated SimCLR reaches performance comparable to the fully supervised baseline with $\geq 50\%$ labelled training data.

## 5   DISCUSSION

In this work we analyze contrastive learning and SimCLR in the federated setting. By adopting a multi-view MI view, we arrive at several interesting observations and extensions. We show that a naive application of local SimCLR training at each client coupled with parameter averaging at the server, corresponds to maximizing a lower bound to the client conditional MI between the two views. We then identify that, in order to close the gap, for global MI an auxiliary user-verification task is necessary. Finally, through the same MI lens, we extend both local and federated SimCLR to the semi-supervised setting in order to handle the case of partially available data. Despite the fact that these modifications were developed through the MI view for SimCLR, we show that they are generally useful for pretraining in the federated setting, yielding improvements for both spectral contrastive learning and SimSiam.

As non-i.i.d. data are an inherent challenge in FL, we further discuss how it affects contrastive learning, both theoretically and empirically. In the case of label skew, the most predominant type of non-i.i.d.-ness in the FL literature, we show that maximizing the global MI through federated SimCLR is appropriate, as the auxiliary user classification task is a good proxy for the unavailable label. On the flipside, in the case of covariate shift, local SimCLR leads to better models due to not being forced to encode irrelevant, for the downstream task, information in the representations.

For future work, we will explore improved variants of the UV loss that can tolerate more local optimization, as well as better bounds for the MI in the federated setting.

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

## A  Experimental setup

**Data partitioning and non-i.i.d.-ness**  For the label-skew setting, we use the Dirichlet splits for CIFAR 10, 100 discussed at Reddi et al. (2020) with $\alpha = 0.1$ in both cases. Notice that we adopt the convention of Hsu et al. (2019) where $\alpha$ is multiplied by the prior probability of the label in the dataset, so, for example, in the case of CIFAR 10 the final concentration parameter is $0.01$.

For the covariate shift setting we consider the case of rotation non-i.i.d.-ness. More specifically, we first perform an i.i.d., with respect to the labels, split of the data into 100 and 500 clients for CIFAR 10 and CIFAR 100 respectively. Afterwards, we bin the $[0, 2\pi]$ range into 10 rotation bins and then assign to each client bins according to a Dirichlet distribution with $\alpha = 0.1$. In this case, each client receives one or two bins of rotations. After that bin assignment, each client randomly rotates each image of their local dataset once with an angle for each image sampled i.i.d. from the bins selected at that client. For the evaluation we consider non-rotated images.

For the joint shift setting we mix the two cases above by first performing a non-i.i.d., Dirichlet split, *i.e.*, $\alpha = 0.1$, according to the labels and then apply the non-i.i.d. rotation strategy described above.

**Architecture details**  For all methods we use the same encoder model, a ResNet18 architecture adapted for CIFAR 10/100 by replacing the kernel of the first convolutional layer with a $3 \times 64 \times 3 \times 3$ kernel and removing the max-pooling and last fully connected layer. Furthermore, to better accommodate for the non-i.i.d. issues in the federated learning scenario (Hsieh et al., 2020) we replace batch normalization (Ioffe & Szegedy, 2015) with group normalization (Wu & He, 2018). For the client ID projector, we use a simple MLP on top of the encoder output with a single ReLU hidden layer of 2048 units and 128 output units. For the auxiliary classifier in the case of semi-supervised learning we use a simple linear layer on top of the encoder output.

For our SimCLR and spectral constrastive learning variants, the representations of the encoder are passed through an MLP projector with a single hidden layer of 2048 units and 128 dimensional outputs. The contrastive loss between the two views is measured at the output of the projector.

For our SimSiam baseline we measure the cosine similarity objective on the output of a projector that follows the SimCLR design with the exception that we also add a group normalization layer before the hidden layer, as SimSiam was unstable without it (especially at the unsupervised experiments). For the predictor we use another single hidden layer MLP with 2048 ReLU units and group normalization.

For the data augmentations, in order to create the two views, we follow the standard recipe of random cropping into $32 \times 32$ images, followed by a random horizontal flip, a random, with probability $0.8$, color distortion with brightness, contrast, saturation factors of $0.4$ and a hue factor of $0.1$. The final augmentation is a random, with probability $0.2$, RGB-to-grayscale transformation.

**Optimization details**  For local optimization we use standard stochastic gradient descent with a learning rate of $0.1$ for both CIFAR 10 and CIFAR 100 for, unless mentioned otherwise, a single local epoch and a batch size of $128$. After the local optimization on a specific round has been completed, each client communicates to the server the delta between the finetuned parameters and the model communicated from the server to the clients. The server averages these deltas, interprets them as "gradients", and uses them in conjunction with the Adam Kingma & Ba (2014) optimizer in order to update the global model. This is a strategy originally proposed in Reddi et al. (2020). For the server-side Adam we are using the default hyperparameters.

## B  Additional experiments

In this section we consider more baselines for both our unsupervised and semi-supervised setups in the federated setting.

### B.1  Unsupervised setting

**Additional baseline**  We consider one more baseline for self-supervised learning in the federated setting, FeatARC (Wang et al., 2022), specifically the "Align Only" variant. We omit the clustering approach as it makes additional assumptions compared to our unsupervised learning setup. The

authors report results with a loss-coefficient of $\lambda = 1.0$, which lead to loss divergence in our case, so we report $\lambda = 0.5$, which was stable except for the covariate shift setting. We see in table 3 that adding FeatARC alignment regularization does not result in improved accuracy, contrary to what the FeatARC paper results would lead us to expect. We hypothesise that this is due to the differences in our setup. Whereas FeatARC considers a cross-silo setting with a large number of local update steps, our setting focuses on the cross-device setting with one local epoch per client communication round. We leave a further analysis of FeatARC applicability to this cross-device setting to future work.

Table 3: Test set performance on the unsupervised setting of CIFAR 10. Clients' data is assumed to be fully annotated for LP fine-tuning in the unsupervised case.

| Method | Label skew | Covariate shift | Joint shift |
|---|---|---|---|
| Local SimCLR | $79.4_{\pm 0.2}$ | $\mathbf{74.3_{\pm 0.3}}$ | $71.0_{\pm 0.4}$ |
| Local SimCLR + FeatARC | $70.4_{\pm 0.2}$ | $34.4_{\pm -}$ | $57.6_{\pm 2.7}$ |
| Federated SimCLR | $\mathbf{85.0_{\pm 0.2}}$ | $73.8_{\pm 0.2}$ | $\mathbf{74.8_{\pm 0.5}}$ |
| Spectral CL | $76.5_{\pm 1.1}$ | $\mathbf{73.5_{\pm 0.4}}$ | $68.2_{\pm 0.6}$ |
| Spectral CL + UV | $\mathbf{87.8_{\pm 0.3}}$ | $71.7_{\pm 0.5}$ | $\mathbf{76.6_{\pm 0.6}}$ |
| SimSiam | $\mathbf{40.0_{\pm 0.5}}$ | $\mathbf{39.9_{\pm 0.3}}$ | $\mathbf{39.6_{\pm 0.3}}$ |
| SimSiam + UV | $35.4_{\pm 0.4}$ | $35.4_{\pm 0.2}$ | $34.5_{\pm 0.3}$ |
| Supervised | $89.6_{\pm 0.1}$ | $78.3_{\pm 0.4}$ | $76.3_{\pm 1.1}$ |

**TinyImagenet dataset** To demonstrate the scalability of our theoretical results and model design stemming from our MI perspective, we also consider the more challenging task of self-supervised pretraining on TinyImagenet. It consists of 100k training examples and 10k test examples, each beloging to one of 200 classes. We apply our federated CIFAR 10 setting to this dataset as well, i.e., we partition the training dataset to 100 clients with either the covariate shift or joint shift non-i.i.d. strategies. We sample 10 clients per round in order to optimize the models and each client performs one local epoch of updates. The encoder mdoel we use is a Compact Convolutional Transformer Hassani et al. (2021) in the "CCT-4/3×2" variant, i.e. with 4 transformer encoder layers and a 2-layer convolutional feature extractor with a 3x3 kernel size. The results with the different methods can be seen at table 4.

Table 4: Test set performance (%) on the unsupervised setting of TinyImagenet with 100 clients after 50k rounds. Clients' data is assumed to be fully annotated for LP fine-tuning in the unsupervised case.

| Method | Label skew | Covariate shift | Joint shift |
|---|---|---|---|
| Local SimCLR | 33.3 | **30.3** | 29.6 |
| Federated SimCLR | **38.0** | 30.0 | **31.6** |
| Spectral CL | 34.0 | 28.4 | 27.9 |
| Spectral CL + UV | **39.7** | **29.5** | **32.4** |
| SimSiam | **10.6** | **4.7** | 0.5 |
| SimSiam + UV | 0.5 | 0.5 | 0.5 |
| Supervised | 44 | 36.6 | 33.0 |

Overall, we see that the results are consistent with our intuitions and story in the case of contrastive methods; the biggest gains from the additional UV loss are in the case of label skew and joint shift. SimSiam generally underperformed in this setting, which is also consistent with our observations in the case of unsupervised learning on CIFAR 10/100, probably due to representation collapse, given that in our setting we use group normalization instead of batch normalization.

### B.2 SEMI-SUPERVISED SETTING

**Additional pseudo-labelling baselines**  We provide more results on our partially labeled (with $10\%$ labeled data on each client) semi-supervised setting by also considering baselines that perform pseudo-labelling as a means for semi-supervised learning. The two methods we consider are SemiFed (Lin et al., 2021) and CBAFed (Li et al., 2023b). For both of these settings we have the following modifications that bring them in line with our semi-supervised setup.

For SemiFed we do not make use of an ensemble of client models in order to impute the missing labels but rather assign a pseudo-label to the datapoint based on the received server model on each client. In this way, our proposed methods and SemiFed have similar communication costs and privacy, as exchanging models directly trained on local data between clients reduces the overall privacy. For CBAFed, we do not use residual weight connection, in order to have a consistent optimization strategy for all our methods, but do use the class balanced adaptive threshold strategy. We follow the setup described in Appendix F.5 of (Li et al., 2023b) to train a model with partially labeled clients.

From what we can see it table 5 and table 6, our conclusion about the usefulness of the UV loss (c.f. proposition 2) applies to this setting as well. While SemiFed underperforms when trained without the UV loss, it manages to improve upon the fully supervised baseline and be comparable to the other methods when we add it back. On CIFAR 10, adding the UV loss yields a significant $16.7\%$ improvement in the case of label skew and on CIFAR 100, while it gets a more modest $6\%$ improvement, it manages to outperform all other methods. CBAFed performs worse than self-supervised methods albeit also benefits from adding the UV loss in all the conducted experiments.

Table 5: Test set performance ($\%$) on the semi-supervised setting of CIFAR 10 with $10\%$ labelled data on each client along with standard error over 5 seeds for all experiments except of CBAFed which have one seed only. We use the corresponding labelled subset for the LP.

| Method | Label skew | Covariate shift | Joint shift |
|---|---|---|---|
| Local SimCLR | $74.5_{\pm 0.3}$ | $\mathbf{49.1_{\pm 1.3}}$ | $45.8_{\pm 1.4}$ |
| Federated SimCLR | $\mathbf{78.0_{\pm 0.2}}$ | $\mathbf{50.3_{\pm 1.1}}$ | $\mathbf{49.9_{\pm 1.4}}$ |
| Spectral CL | $74.2_{\pm 0.3}$ | $48.0_{\pm 0.7}$ | $45.4_{\pm 1.5}$ |
| Spectral CL + UV | $\mathbf{79.6_{\pm 0.3}}$ | $\mathbf{49.7_{\pm 1.0}}$ | $\mathbf{49.8_{\pm 1.1}}$ |
| SimSiam | $75.3_{\pm 0.4}$ | $46.8_{\pm 0.7}$ | $40.5_{\pm 0.9}$ |
| SimSiam + UV | $\mathbf{80.4_{\pm 0.2}}$ | $\mathbf{50.0_{\pm 1.2}}$ | $\mathbf{44.3_{\pm 1.0}}$ |
| SemiFed | $60.0_{\pm 4.5}$ | $18.6_{\pm 1.8}$ | $37.2_{\pm 0.9}$ |
| SemiFed + UV | $\mathbf{76.7_{\pm 1.2}}$ | $\mathbf{24.0_{\pm 2.2}}$ | $\mathbf{45.1_{\pm 2.0}}$ |
| CBAFed | $66.3$ | $45.9$ | $34.8$ |
| CBAFed + UV | $\mathbf{74.1}$ | $\mathbf{48.2}$ | $\mathbf{36.2}$ |
| Supervised | $75.1_{\pm 0.2}$ | $48.1_{\pm 0.9}$ | $42.7_{\pm 1.7}$ |

**TinyImagenet dataset**  To demonstrate the scalability of our semi-supervised model design stemming from our MI perspective, we also consider the more challenging TinyImagenet task in the case of label skew non-i.i.d.-ness with Dirichlet splitting and an $\alpha = 0.1$ multiplied by the prior probability of each class. The setup is similar to our semi-supervised federated CIFAR 10 setting, with 100 clients and $10\%$ labelled data per client. We sample 10 clients per round in order to optimize the models and each client performs one local epoch of updates. We use the same CCT architecture as the unsupervised TinyImagenet experiment. The results with the different methods can be seen in table 7.

We observe similar patterns to our unsupervised TinyImagenet setting, with the biggest gains for the contrastive methods from the UV loss being in the case where some label skew is present. SimSiam did experience representation collapse at the case of label skew, however, by adding to it the UV loss, this was successfully mitigated and improved significantly the performance.

Table 6: Test set performance (%) on the semi-supervised setting of CIFAR 100 with 10% labelled data on each client along with standard error over 5 seeds. We use the corresponding labelled subset for the LP.

| Method | Label Skew | Covariate shift | Joint shift |
|---|---|---|---|
| Local SimCLR | $30.3_{\pm 0.2}$ | $15.1_{\pm 0.4}$ | $13.1_{\pm 0.3}$ |
| Federated SimCLR | $\mathbf{34.5_{\pm 0.3}}$ | $14.8_{\pm 0.3}$ | $\mathbf{14.6_{\pm 0.3}}$ |
| Spectral CL | $30.1_{\pm 0.2}$ | $14.1_{\pm 0.4}$ | $12.3_{\pm 0.3}$ |
| Spectral CL + UV | $\mathbf{34.0_{\pm 0.2}}$ | $13.7_{\pm 0.3}$ | $\mathbf{13.6_{\pm 0.4}}$ |
| SimSiam | $30.7_{\pm 0.2}$ | $13.4_{\pm 0.3}$ | $12.8_{\pm 0.3}$ |
| SimSiam + UV | $\mathbf{34.3_{\pm 0.1}}$ | $13.6_{\pm 0.3}$ | $\mathbf{14.0_{\pm 0.4}}$ |
| SemiFed | $29.7_{\pm 0.5}$ | $13.3_{\pm 0.2}$ | $12.3_{\pm 0.2}$ |
| SemiFed + UV | $\mathbf{35.7_{\pm 0.2}}$ | $13.4_{\pm 0.6}$ | $\mathbf{13.1_{\pm 0.2}}$ |
| Supervised | $29.6_{\pm 0.3}$ | $12.6_{\pm 0.2}$ | $12.2_{\pm 0.1}$ |

Table 7: Test set performance (%) on the semi-supervised setting of TinyImagenet with 100 clients after 50k rounds. We use the corresponding labelled subset for the linear probe.

| Method | Label skew | Covariate shift | Joint shift |
|---|---|---|---|
| Local SimCLR | 18.5 | 8.1 | 6.7 |
| Federated SimCLR | **19.5** | **8.4** | **7.4** |
| Spectral CL | 17.8 | **8.3** | 6.9 |
| Spectral CL + UV | **18.9** | 8.1 | **7.5** |
| SimSiam | 0.5 | 8.1 | **6.9** |
| SimSiam + UV | **20.0** | **8.5** | 6.9 |
| Supervised | 17.9 | 8.4 | 7.7 |

## C    ALGORITHMS

---

**Algorithm 1** The server side algorithm for our federated SimCLR / Spectral CL / SimSiam with optional user-verification and semi-supervision.

---

Initialize $\theta$ and $\phi$ with $\theta_1, \phi_i$
**for** round $t$ in $1, \ldots T$ **do**
    Sample $\mathcal{S}$ clients from the population
    Initialize $\nabla_\theta^t = \mathbf{0}, \nabla_\phi^t = \mathbf{0}$
    **for** $s$ in $\mathcal{S}$ **do**
        $\theta_s, \phi_s \leftarrow \textsc{Client}(s, \theta_t, \phi_t)$
        $\nabla_\theta^t += \frac{\theta_t - \theta_s}{|\mathcal{S}|}$
        $\nabla_\phi^t += \frac{\phi_t - \phi_s}{|\mathcal{S}|}$
    **end for**
    $\theta^{t+1}, \phi^{t+1} \leftarrow \textsc{Adam}(\nabla_\theta^t, \nabla_\phi^t)$
**end for**

---

---

**Algorithm 2** The client side algorithm for our federated SimCLR / Spectral CL / SimSiam with optional user-verification and semi-supervision. $L_{ul}$ corresponds to the unsupervised loss component of SimCLR / Spectral CL / SimSiam. $\beta$ is a coefficient that determines the weight of the UV loss, with a default value of 1.

---

Get $\theta, \phi$ from the server
$\theta_s, \phi_s \leftarrow \theta, \phi$
**for** epoch $e$ in $1, \dots, E$ **do**
    **for** batch $b \in B$ **do**
        $\triangleright$ Unlabelled and labelled datapoints of the batch $b$
        $x_{ul}, (x_l, y_l) \leftarrow b$
        $\triangleright$ Get the two views through augmentations
        $[x_{ul}^1, x_l^1], [x_{ul}^2, x_l^2] = \text{AUG}([x_{ul}, x_l]), \text{AUG}([x_{ul}, x_l])$
        $\triangleright$ Representations of the two views from the encoder $f$ with parameters $\theta_s$
        $[z_{ul}^1, z_l^1], [z_{ul}^2, z_l^2] \leftarrow f([x_{ul}^1, x_l^1]; \theta_s), f([x_{ul}^2, x_l^2]; \theta_s)$
        $\triangleright$ Unsupervised loss with, depending on $\beta$, an additional UV loss
        $\mathcal{L}_s = \mathcal{L}_{ul}(z_{ul}^1, z_{ul}^2; \phi_s) + \beta \mathcal{L}_{uv}(s, z_{ul}^1, z_{ul}^2; \phi_s)$
        $\triangleright$ Supervised loss on the labelled data
        **for** label $i \in \{0, \dots, |Y| - 1\}$ **do**
            $\triangleright$ Unsupervised loss between datapoints of the same class
            $\mathcal{L}_s += \mathcal{L}_{ul}(z_l^1[y_l == i], z_l^2[y_l == i]; \phi_s)$
        **end for**
        $\triangleright$ Standard supervised loss
        $\mathcal{L}_s += \mathcal{L}_y(y_l, z_l^1, z_l^2; \phi_s)$
        $\triangleright$ Local gradient updates on the loss
        $\theta_s, \phi_s \leftarrow \text{SGD}(\nabla_{\theta_s, \phi_s} L_s)$
    **end for**
**end for**
**return** $\theta_s, \phi_s$

---

## D  MISSING PROOFS

**Proposition 1.** *Let* $s \in \mathbb{N}$ *denote the user ID,* $\mathbf{x} \in \mathbb{R}^{D_x}$ *the input and* $\mathbf{z}_1, \mathbf{z}_2 \in \mathbb{R}^{D_z}$ *the latent representations of the two views of* $\mathbf{x}$ *given by the encoder with parameters* $\theta$. *Given a critic function* $f : \mathbb{R}^{D_z} \times \mathbb{R}^{D_z} \to \mathbb{R}$, *we have that*

$$\mathrm{I}_\theta(\mathbf{z}_1; \mathbf{z}_2 | s) \geq \mathbb{E}_{p(s)p_\theta(\mathbf{z}_1, \mathbf{z}_2|s)_{1:K}} \left[ \frac{1}{K} \sum_{k=1}^K \log \frac{\exp(f(\mathbf{z}_{1k}, \mathbf{z}_{2k}))}{\frac{1}{K} \sum_{j=1}^K \exp(f(\mathbf{z}_{1j}, \mathbf{z}_{2k}))} \right]. \tag{14}$$

*Proof.* The proof follows Poole et al. (2019). We can show that

$$I_\theta(\mathbf{z}_1; \mathbf{z}_2|s) = \mathbb{E}_{p(s)p_\theta(\mathbf{z}_{1,1}, \mathbf{z}_2|s)p_\theta(\mathbf{z}_{1,2:K}|s)} \left[ \log \frac{p_\theta(\mathbf{z}_{1,1}|\mathbf{z}_2, s)p_\theta(\mathbf{z}_{1,2:K}|s)}{p_\theta(\mathbf{z}_{1,2:K}|s)p_\theta(\mathbf{z}_{1,1}|s)} \right] \tag{15}$$

$$= \mathbb{E}_{p(s)p_\theta(\mathbf{z}_{1,1:K}, \mathbf{z}_2|s)} \left[ \log \frac{p_\theta(\mathbf{z}_{1,1:K}|\mathbf{z}_2, s)}{p_\theta(\mathbf{z}_{1,1:K}|s)} \right] \tag{16}$$

$$= \mathbb{E}_{p(s)p_\theta(\mathbf{z}_{1,1:K}, \mathbf{z}_2|s)} \left[ \log \frac{p_\theta(\mathbf{z}_{1,1:K}|\mathbf{z}_2, s)q(\mathbf{z}_{1,1:K}|\mathbf{z}_2, s)}{q(\mathbf{z}_{1,1:K}|\mathbf{z}_2, s)p_\theta(\mathbf{z}_{1,1:K}|s)} \right] \tag{17}$$

$$= \mathbb{E}_{p(s)p_\theta(\mathbf{z}_{1,1:K}, \mathbf{z}_2|s)} \left[ \log \frac{q(\mathbf{z}_{1,1:K}|\mathbf{z}_2, s)}{p_\theta(\mathbf{z}_{1,1:K}|s)} \right]$$
$$+ \mathbb{E}_{p(s)p_\theta(\mathbf{z}_2|s)p_\theta(\mathbf{z}_{1,1:K}|\mathbf{z}_2, s)} \left[ \log \frac{p_\theta(\mathbf{z}_{1,1:K}|\mathbf{z}_2, s)}{q(\mathbf{z}_{1,1:K}|\mathbf{z}_2, s)} \right] \tag{18}$$

$$= \mathbb{E}_{p(s)p_\theta(\mathbf{z}_{1,1:K}, \mathbf{z}_2|s)} \left[ \log \frac{q(\mathbf{z}_{1,1:K}|\mathbf{z}_2, s)}{p_\theta(\mathbf{z}_{1,1:K}|s)} \right]$$
$$+ \mathbb{E}_{p(s)p_\theta(\mathbf{z}_2|s)} \left[ D_{\mathrm{KL}}(p_\theta(\mathbf{z}_{1,1:K}|\mathbf{z}_2, s)||q(\mathbf{z}_{1,1:K}|\mathbf{z}_2, s)) \right] \tag{19}$$

$$\geq \mathbb{E}_{p(s)p_\theta(\mathbf{z}_{1,1:K}, \mathbf{z}_2|s)} \left[ \log \frac{q(\mathbf{z}_{1,1:K}|\mathbf{z}_2, s)}{p_\theta(\mathbf{z}_{1,1:K}|s)} \right], \tag{20}$$

and then by parametrizing $q(\mathbf{z}_{1,1:K}|\mathbf{z}_2, s)$ in terms of a critic function $f$,

$$q(\mathbf{z}_{1,1:K}|\mathbf{z}_2, s) = \frac{p_\theta(\mathbf{z}_{1,1:K}|s) \exp(f(\mathbf{z}_2, \mathbf{z}_{1,1:K}))}{\mathbb{E}_{p_\theta(\mathbf{z}_{1,1:K}|s)}[\exp(f(\mathbf{z}_2, \mathbf{z}_{1,1:K}))]}, \tag{21}$$

we have that

$$I_\theta(\mathbf{z}_1; \mathbf{z}_2|s) \geq \mathbb{E}_{p(s)p_\theta(\mathbf{z}_{1,1:K}, \mathbf{z}_2|s)} \left[ \log \frac{\exp(f(\mathbf{z}_2, \mathbf{z}_{1,1:K}))}{\mathbb{E}_{p_\theta(\mathbf{z}_{1,1:K}|s)}[\exp(f(\mathbf{z}_2, \mathbf{z}_{1,1:K}))]} \right]. \tag{22}$$

Since the denominator depends on the aggregate score $\exp(f(\mathbf{z}_2, \mathbf{z}_{1,1:K}))$ over $p_\theta(\mathbf{z}_{1,1:K}|s)$, which is similarly intractable, we can introduce one more lower bound that will allow us to work with minibatches of data Poole et al. (2019). Due to the positivity of the exponent, we have that for any $a > 0$

$$\log \mathbb{E}_{p_\theta(\mathbf{z}_{1,1:K}|s)}[\exp(f(\mathbf{z}_2, \mathbf{z}_{1,1:K}))] \leq \frac{\mathbb{E}_{p_\theta(\mathbf{z}_{1,1:K}|s)}[\exp(f(\mathbf{z}_2, \mathbf{z}_{1,1:K}))]}{a} + \log a - 1. \tag{23}$$

Using this bound with $\alpha = \exp(1)$, we have that

$$I_\theta(\mathbf{z}_1; \mathbf{z}_2|s) \geq \mathbb{E}_{p(s)p_\theta(\mathbf{z}_{1:K}, \mathbf{z}_2|s)} \left[ \log \exp(f(\mathbf{z}_2, \mathbf{z}_{1,1:K})) \right]$$
$$- \exp(-1)\mathbb{E}_{p(s)p_\theta(\mathbf{z}_2|s)p_\theta(\mathbf{z}_{1,1:K}|s)} \left[ \exp(f(\mathbf{z}_2, \mathbf{z}_{1,1:K})) \right]. \tag{24}$$

We can now set $f(\mathbf{z}_2, \mathbf{z}_{1,1:K})$ as Poole et al. (2019)

$$f(\mathbf{z}_2, \mathbf{z}_{1,1:K}) \to 1 + f(\mathbf{z}_2, \mathbf{z}_{1,1}) - \log a(\mathbf{z}_2, \mathbf{z}_{1,1:K}). \tag{25}$$

In this way, we end up with

$$I_\theta(\mathbf{z}_1; \mathbf{z}_2|s) \geq 1 + \mathbb{E}_{p(s)p_\theta(\mathbf{z}_2, \mathbf{z}_{1,1:K}|s)} \left[ \log \frac{\exp(f(\mathbf{z}_2, \mathbf{z}_{1,1}))}{a(\mathbf{z}_2, \mathbf{z}_{1,1:K})} \right]$$
$$- \mathbb{E}_{p(s)p_\theta(\mathbf{z}_2|s)p_\theta(\mathbf{z}_{1,1:K}|s)} \left[ \frac{\exp(f(\mathbf{z}_2, \mathbf{z}_{1,1}))}{a(\mathbf{z}_2, \mathbf{z}_{1,1:K})} \right]. \tag{26}$$

We can now average the bound over $K$ replicates and reindex $\mathbf{z}_1$ as

$$
\begin{aligned}
I_\theta(\mathbf{z}_1; \mathbf{z}_2|s) \geq 1 + \frac{1}{K} \sum_{k=1}^{K} & \left( \mathbb{E}_{p(s)p_\theta(\mathbf{z}_2,\mathbf{z}_{1,1:K}|s)} \left[ \log \frac{\exp(f(\mathbf{z}_2, \mathbf{z}_{1,1}))}{a(\mathbf{z}_2, \mathbf{z}_{1,1:K})} \right] \right. \\
& \left. - \mathbb{E}_{p(s)p_\theta(\mathbf{z}_2|s)p_\theta(\mathbf{z}_{1,1:K}|s)} \left[ \frac{\exp(f(\mathbf{z}_2, \mathbf{z}_{1,1}))}{a(\mathbf{z}_2, \mathbf{z}_{1,1:K})} \right] \right)
\end{aligned}
\tag{27}
$$

$$
\begin{aligned}
= 1 + \frac{1}{K} \sum_{k=1}^{K} & \mathbb{E}_{p(s)p_\theta(\mathbf{z}_2,\mathbf{z}_{1,1:K}|s)} \left[ \log \frac{\exp(f(\mathbf{z}_2, \mathbf{z}_{1,1}))}{a(\mathbf{z}_2, \mathbf{z}_{1,1:K})} \right] \\
& - \frac{1}{K} \sum_{k=1}^{K} \mathbb{E}_{p(s)p_\theta(\mathbf{z}_2|s)p_\theta(\mathbf{z}_{1,1:K}|s)} \left[ \frac{\exp(f(\mathbf{z}_2, \mathbf{z}_{1,1}))}{a(\mathbf{z}_2, \mathbf{z}_{1,1:K})} \right]
\end{aligned}
\tag{28}
$$

$$
\begin{aligned}
= 1 + & \mathbb{E}_{p(s)p_\theta(\mathbf{z}_2,\mathbf{z}_{1,1:K}|s)} \left[ \frac{1}{K} \sum_{k=1}^{K} \log \frac{\exp(f(\mathbf{z}_2, \mathbf{z}_{1,k}))}{a(\mathbf{z}_2, \mathbf{z}_{1,1:K})} \right] \\
& - \frac{1}{K} \sum_{k=1}^{K} \mathbb{E}_{p(s)p_\theta(\mathbf{z}_2|s)p_\theta(\mathbf{z}_{1,1:K}|s)} \left[ \frac{\exp(f(\mathbf{z}_2, \mathbf{z}_{1,k}))}{a(\mathbf{z}_2, \mathbf{z}_{1,1:K})} \right]
\end{aligned}
\tag{29}
$$

and for the specific choice of $a(\mathbf{z}_2, \mathbf{z}_{1,1:K}) = \frac{1}{K} \sum_{k=1}^{K} \exp(f(\mathbf{z}_2, \mathbf{z}_{1,k}))$, we have that terms cancel, i.e.,

$$
\frac{1}{K} \sum_{k=1}^{K} \mathbb{E}_{p(s)p_\theta(\mathbf{z}_2|s)p_\theta(\mathbf{z}_{1,1:K}|s)} \left[ \frac{\exp(f(\mathbf{z}_2, \mathbf{z}_{1,k}))}{\frac{1}{K} \sum_{k=1}^{K} \exp(f(\mathbf{z}_2, \mathbf{z}_{1,k}))} \right]
$$

$$
= \mathbb{E}_{p(s)p_\theta(\mathbf{z}_2|s)p_\theta(\mathbf{z}_{1,1:K}|s)} \left[ \frac{\frac{1}{K} \sum_{k=1}^{K} \exp(f(\mathbf{z}_2, \mathbf{z}_{1,k}))}{\frac{1}{K} \sum_{k=1}^{K} \exp(f(\mathbf{z}_2, \mathbf{z}_{1,k}))} \right] = 1.
\tag{30}
$$

In this way, we end up with the well known InfoNCE loss Oord et al. (2018), where now we contrast between datapoints that share the same class

$$
I_\theta(\mathbf{z}_1; \mathbf{z}_2|s) \geq \mathbb{E}_{p(s)p_\theta(\mathbf{z}_1,\mathbf{z}_2|s)_{1:K}} \left[ \frac{1}{K} \sum_{k=1}^{K} \log \frac{\exp(f(\mathbf{z}_{1k}, \mathbf{z}_{2k}))}{\frac{1}{K} \sum_{j=1}^{K} \exp(f(\mathbf{z}_{1j}, \mathbf{z}_{2k}))} \right].
\tag{31}
$$

$\square$

**Lemma 2.1.** *Let $s \in \mathbb{N}$ denote the client ID, $\mathbf{x} \in \mathbb{R}^{D_x}$ the input and $\mathbf{z}_1 \in \mathbb{R}^{D_z}$ the latent representation of a view of $\mathbf{x}$ given by the encoder with parameters $\theta$. Let $\phi$ denote the parameters of a client classifier $r_\phi(s|\mathbf{z}_1)$ that predicts the client ID from this specific representation and let $H(s)$ be the entropy of the client distribution $p(s)$. We have that*

$$
I_\theta(\mathbf{z}_1; s) \geq \mathbb{E}_{p(s)p_\theta(\mathbf{z}_1|s)} [\log r_\phi(s|\mathbf{z}_1)] + H(s)
\tag{32}
$$

*Proof.*

$$
I_\theta(\mathbf{z}_1; s) = \mathbb{E}_{p_\theta(s,\mathbf{z}_1)} \left[ \log \frac{p_\theta(s, \mathbf{z}_1)}{p(s)p_\theta(\mathbf{z}_1)} \right] = \mathbb{E}_{p(s)p_\theta(\mathbf{z}_1|s)} \left[ \log \frac{p_\theta(s|\mathbf{z}_1)}{p(s)} \right]
\tag{33}
$$

$$
= \mathbb{E}_{p(s)p_\theta(\mathbf{z}_1|s)} \left[ \log \frac{r_\phi(s|\mathbf{z}_1)}{p(s)} \right] + \mathbb{E}_{p(s)} [D_{\mathrm{KL}}(p_\theta(s|\mathbf{z}_1)||r_\phi(s|\mathbf{z}_1))]
\tag{34}
$$

$$
\geq \mathbb{E}_{p(s)p_\theta(\mathbf{z}_1|s)} \left[ \log \frac{r_\phi(s|\mathbf{z}_1)}{p(s)} \right] = \mathbb{E}_{p(s)p_\theta(\mathbf{z}_1|s)} [\log r_\phi(s|\mathbf{z}_1)] + H(s).
\tag{35}
$$

$\square$

**Lemma 2.2.** *Let $s \in \mathbb{N}$ denote the user ID, $\mathbf{x} \in \mathbb{R}^{D_x}$ the input and $\mathbf{z}_1, \mathbf{z}_2 \in \mathbb{R}^{D_z}$ the latent representations of the views of $\mathbf{x}$ given by the encoder with parameters $\theta$. Let $\phi$ denote the parameters of a client classifier $r_\phi(s|\mathbf{z}_2)$ that predicts the client ID from the representations. We have that*

$$
I_\theta(\mathbf{z}_1; s|\mathbf{z}_2) \leq -\mathbb{E}_{p(s)p_\theta(\mathbf{z}_2|s)} [\log r_\phi(s|\mathbf{z}_2)]
\tag{36}
$$

*Proof.*

$$I_\theta(\mathbf{z}_1; s|\mathbf{z}_2) = H_\theta(s|\mathbf{z}_2) - H_\theta(s|\mathbf{z}_2, \mathbf{z}_1) \tag{37}$$

$$\leq H_\theta(s|\mathbf{z}_2) = H(s) - I_\theta(\mathbf{z}_2; s) \leq -\mathbb{E}_{p(s)p_\theta(\mathbf{z}_2|s)}\left[\log r_\phi(s|\mathbf{z}_2)\right] \tag{38}$$

where $H(s)$ is the entropy of $p(s)$, $H_\theta(s|\mathbf{z}_2)$, $H_\theta(s|\mathbf{z}_2, \mathbf{z}_1)$ are the conditional entropies of $s$ given $\mathbf{z}_2$ and $\mathbf{z}_2, \mathbf{z}_1$ and the last inequality is due to the lower bound of lemma 2.1. We also used the fact that the entropy of a discrete distribution is non-negative. $\square$

**Proposition 2**. *Consider the label skew data-generating process for federated SimCLR from Figure 1 with $s \in \mathbb{N}$ denoting the user ID with $H(s)$ being the entropy of $p(s)$, $\mathbf{x} \in \mathbb{R}^{D_x}$ the input, $\mathbf{z}_1, \mathbf{z}_2 \in \mathbb{R}^{D_z}$ the latent representations of the two views of $\mathbf{x}$ given by the encoder with parameters $\theta$. Let $y$ be the label and let $r_\phi(s|\mathbf{z}_i)$ be a model with parameters $\phi$ that predicts the user ID from the latent representation $\mathbf{z}_i$. In this case, we have that*

$$I_\theta(\mathbf{z}_1; y) + I_\theta(\mathbf{z}_2; y) \geq \mathbb{E}_{p(s)p_\theta(\mathbf{z}_1, \mathbf{z}_2|s)}\left[\log r_\phi(s|\mathbf{z}_1) + \log r_\phi(s|\mathbf{z}_2)\right] + 2H(s). \tag{39}$$

*Proof.* The claim is a consequence of the data processing inequality. We start by noting that

$$I_\theta(\mathbf{z}_1; y) + I_\theta(\mathbf{z}_1; s|y) = I_\theta(\mathbf{z}_1; y, s) = I_\theta(\mathbf{z}_1; s) + I_\theta(\mathbf{z}_1; y|s) \tag{40}$$

and since in this graphical model we have that $s \perp\!\!\!\perp \mathbf{z}_1|y$, so $I_\theta(s; \mathbf{z}_1|y) = 0$, we end up with

$$I_\theta(\mathbf{z}_1; y) = I_\theta(\mathbf{z}_1; s) + I_\theta(\mathbf{z}_1; y|s) \tag{41}$$

$$\geq I_\theta(\mathbf{z}_1; s) \geq \mathbb{E}_{p(s)p_\theta(\mathbf{z}_1|s)}\left[\log r_\phi(s|\mathbf{z}_1)\right] + H(s), \tag{42}$$

where we use the positivity of mutual information and our lemma 2.1. In a similar manner we can also show that

$$I_\theta(\mathbf{z}_2; y) \geq \mathbb{E}_{p(s)p_\theta(\mathbf{z}_2|s)}\left[\log r_\phi(s|\mathbf{z}_2)\right] + H(s). \tag{43}$$

By adding up eq. (42) and eq. (43) we arrive at the claim. $\square$

