# OpenReview forum: "A Mutual Information Perspective on Federated Contrastive Learning"
_ICLR.cc/2024/Conference — ICLR 2024 spotlight_

### Official Review · Reviewer_3B5H · 2023-10-31

**Soundness:** 3 good
**Presentation:** 3 good
**Contribution:** 2 fair
**Rating:** 5
**Confidence:** 4

**Summary:**

The paper presents an extension of SimCLR, a contrastive learning framework, to the federated learning (FL) setting, with a focus on mutual information maximization (MI) across multiple views.
The authors establish a connection between contrastive representation learning and user verification and propose a method that incorporates a user verification loss into each client’s local SimCLR loss, resulting in a lower bound to the global multi-view mutual information.
Additionally, the paper extends the approach to the federated semi-supervised setting, introducing modifications to accommodate labelled data at the clients and proposing an auxiliary head for label prediction. The paper also investigates the impact of different sources of non-i.i.d. data distribution on federated unsupervised learning performance.

**Strengths:**

The extension of SimCLR to the federated setting and the exploration of MI maximization in this context is particularly given the increasing interest in FL.
The paper provides a theoretical foundation for the proposed methods, including the connection between contrastive learning and user verification, and the derivation of a lower bound to the global multi-view MI.
The authors conduct both unsupervised and semi-supervised experiments, providing a thorough evaluation of their proposed method.

**Weaknesses:**

The theoretical derivations, propositions, and lemmas mainly connect to and extend existing methods, which might be perceived as a lack of novelty. For example, the idea of decomposed MI in (1) and (2) has been presented in Sordoni et al. (2021). The proofs of propositions and lemmas are quite standard which mostly follow the existing approaches in literature. It would be more beneficial if the authors can clarify the unique aspects and advantages of their approach, and clearly differentiate it from existing methods.
Furthermore, the authors only provide analysis for a two-view setting, which might not be completely satisfied with the proposed multi-view MI.
The paper does not present algorithms for federated training, which is crucial for practical implementation. Moreover, as a federated learning algorithm, there should be a thorough analysis of the convergence guarantees, which seems to be lacking.
The experimental setup presented in the paper demonstrates a certain degree of comprehensiveness; however, it appears to be somewhat limited in terms of diversity. The FL baselines utilized in the study are mainly adaptations from centralized methods, which may not fully represent the state-of-the-art in unsupervised representation learning within the FL context.
Looking at the results outlined in Tables 1 and 2, it becomes evident that in a majority of the scenarios, the performance of the proposed method is either on par with or falls short of other unsupervised baselines. This observation raises questions about the clear and tangible benefits of the proposed approach.

**Questions:**

Additionally, the proposed multi-view MI estimation might result in additional computation overhead.  This is particularly crucial in FL where computational resources are may be limited. Therefore, the paper could be significantly enhanced by a more thorough analysis and discussion of the trade-offs between performance and computational complexity.

---

> ### Author Response · Authors · 2023-11-17
> **Response to review**
>
> We thank you for the thoughtful review and the encouraging comments about our work. We hope that with the shared response and the following we address your concerns:
>
> -	*Theory*: Indeed, Sordoni et al. has also proposed a decomposition of the MI, but we argue that the main goal of Sordoni et al. is different than ours. Sordoni et al. decomposes the MI to get a better estimation of the MI with InfoNCE type of bounds, whereas in our case the decomposition happens due to the partitioning of the data in the federated setting. As a result, we believe that our contribution is orthogonal, since, e.g., the methods from Sordoni et al. could be applied for the SimCLR training on each client. Overall, we believe that the decomposition of the MI itself is not our novel result per se, as it is a trivial consequence of the mutual information properties. We simply highlight this decomposition in order to show that a) simple SimCLR on each client (a common baseline in federated SSL [1, 2]) optimizes one term of the decomposition and b)  (which is one of our main results) how through that decomposition and an auxiliary user-verification task we can lower bound the global mutual information. Furthermore, through this (similar) decomposition we were able to propose a novel, from the perspective of MI, extension of SimCLR (and our baselines) to the semi-supervised setting. We will update the paper accordingly.
>
> -	*Amount of views*: we use only two views as we adopt the standard convention for SSL methods.
>
> -	*Algorithms for federated training*: indeed, this is a good point and we updated the paper to include algorithms in the appendix.
>
> -	*Performance relative to baselines*: please also see the response to wQ6s; overall **the baselines do sometimes better only after we incorporate the intuitions and theoretical results from our MI and FedSimCLR discussion.**
>
> -	*Computational overhead*: overall, we, at most, include just two additional heads to the feature encoder which, compared to the feature encoder itself, do not add have much extra computational overhead. We have revised the paper accordingly.
>
> We hope that with the shared response and the above we address your concerns, and we are happy to continue the discussion otherwise.
>
> [1] Federated Unsupervised Representation Learning, Zhang et al., 2020
>
> [2] Federated Contrastive Learning for Decentralized Unlabeled Medical Images, 2021

---

> ### Author Response · Authors · 2023-11-22
> **Last day of discussion**
>
> Dear 3B5H,
>
> The discussion period is nearing its end, so we would like to ask you whether our rebuttal and revised submission adequately address your concerns.  More specifically, per your request, we have done the following changes in the submission
>
> - We have discussed Sordoni et al.
> - We have added in the appendix the algorithms for federated training with our methods
> - We have discussed about the convergence guarantees of our methods
> - We have increased the diversity of our experiments by considering more baselines in both the unsupervised and semi-supervised setting.
> - We have updated the discussion to better highlight that the baselines do better only **after** we incorporate the intuitions from our theory and mutual information perspective.
> - We have discussed the computational overhead of our methods.
>
> We are happy to engage further in discussions to clarify any confusion or misunderstanding.

---

> > ### Comment · Reviewer_3B5H · 2023-12-04
> > **response to the rebuttal**
> >
> > I thank the authors for addressing my comments, and sorry for the delay due to my travel.
> > Among all of the points discussed, I still have concerns about the federated training algorithms provided in the appendix. First, two Algorithms 1 and 2 are provided without any explanation. For example, what are the meanings and details of sub-routines CLIENTS and ADAM in Algorithm 1, and CONSTRAINT in Algorithm 2? Second, I'm not sure if the following update is related to local model updates of clients as in standard FL.
> > \begin{equation}
> > \nabla^t_\theta + =\frac{\theta_t - \theta_s}{|S|} \qquad \nabla^t_\phi += \frac{\phi_t - \phi_s}{|S|}
> > \end{equation}
> > Finally, I'm not sure the convergence is guaranteed because there's CONSTRAINT in Algorithm 2, which makes it a constraint problem, which is different from the referenced paper
> > "Sashank Reddi, Zachary Charles, Manzil Zaheer, Zachary Garrett, Keith Rush, Jakub Konecnˇ y, `
> > Sanjiv Kumar, and H Brendan McMahan. Adaptive federated optimization. arXiv preprint
> > arXiv:2003.00295, 2020."
> >
> > Additionally, about the general multi-view SSL (instead of only 2-view SSL), by a quick Googling I can find a relevant paper: https://arxiv.org/pdf/2307.09614.pdf

---

### Official Review · Reviewer_wQ6s · 2023-10-31

**Soundness:** 3 good
**Presentation:** 3 good
**Contribution:** 2 fair
**Rating:** 8
**Confidence:** 4

**Summary:**

The paper presents a federated variant of SimCLR for unsupervised representation learning. It motivates its approach by a mutual information argument: since in standard SimCLR the goal is to maximize the mutual information (MI) between the two generated views, this MI can be decomposed into a local variant that corresponds to local SimCLR and two excess terms that need to be bounded. The first relates the mutual information between the first view and its local client, which is lower bounded using a classifier that seeks to predict the client ID from the first view. The second term relates to the additional or excess mutual information of the second view on the client which can be upper bounded by a second classifier that predicts the client ID from the second view. In addition, the paper presents a semi-supervised variant of this approach.

**Strengths:**

- unsupervised federated representation learning is an important and interesting use-case
- the method theoretically motivated and sound

**Weaknesses:**

- some baselines for semi-supervised learning, and some proper supervised baselines are missing.
- The empirical results show that the proposed federated SimCLR variant is only en par with spectral constrastive learning when using a user verification loss. This is not properly discussed.

**Questions:**

**Question:**
- How does the semi-supervised variant of SimCLR relate to pseudo-labeling approaches, such as distributed distillation [2] and federated co-training [1]?

**Detailed Comments:**

- Please discuss the empirical results, in particular the fact that FedSimCLR is not outperforming the baselines, in more detail. The proposed method does not have to outperform the baselines, as long as the benefits and limitations of it in comparison with existing methods are properly discussed. To stress this point: it has become usual to require papers to have large tables where the proposed method has "the best number" in each row, but this just promotes scientifically questionable practices to improve the numbers. I am happy that this paper presents more interesting results, but they require, unfortunately, a more thorough discussion. One could even, for example, use mutual information as quality measure (where one would probably rely on the more tractable Wasserstein dependency measure [6], isntead of approximating MI).
- Please state what exactly the supervised baseline in your experiments is (I assume FedAvg). Please compare to (one of the) FL variants for non-iid data, such as FedProx [4], FedBN [5], and SCAFFOLD [3], as baselines.
- For the semi-supervised setting, please compare to pseudo-labeling approaches [1,2] as semi-supervised baselines.


[1] Abourayya, Amr, et al. "Protecting Sensitive Data through Federated Co-Training." arXiv preprint arXiv:2310.05696 (2023).\
[2] Bistritz, Ilai, Ariana Mann, and Nicholas Bambos. "Distributed distillation for on-device learning." Advances in Neural Information Processing Systems 33 (2020): 22593-22604.\
[3] Karimireddy, Sai Praneeth, et al. "Scaffold: Stochastic controlled averaging for federated learning." International conference on machine learning. PMLR, 2020.\
[4] Li, Tian, et al. "Federated optimization in heterogeneous networks." Proceedings of Machine learning and systems 2 (2020): 429-450.\
[5] Li, Xiaoxiao, et al. "FedBN: Federated Learning on Non-IID Features via Local Batch Normalization." International Conference on Learning Representations. 2021.\
[6] Ozair, Sherjil, et al. "Wasserstein dependency measure for representation learning." Advances in Neural Information Processing Systems 32 (2019).

---

> ### Author Response · Authors · 2023-11-17
> **Response to review**
>
> We thank you for the insightful comments and for acknowledging the theoretical motivation and soundness of our work. We appreciate that you find our results interesting and, together with the shared response, we will provide more details in support of our work
>
> -	*Empirical results*: we consider FedSimCLR as one instance of a method that optimizes a global pretraining objective, stemming from the following intuition. Since on each client the model contrasts between datapoints of the same distribution, a way to close the gap to the centralized model training is by having the model also distinguishing between users as well, i.e., a user-verification objective. In the case of mutual information this becomes precise due to the user-verification task acting as a lower bound to the gap between the global and local MI. Furthermore, we also demonstrate theoretically at Proposition 3 that closing the gap with such an auxiliary objective is beneficial, as in itself is a lower bound to the, unavailable, label  classification objective. Based on these intuitions from our perspective and theory, we do extend both Spectral CL and SimSiam in a similar manner (i.e., Spectral CL + UV and SimSiam +UV) and we do observe that the results generalize similarly. However, **Spectral CL does better only when incorporating the results from our theory and is worse than FedSimCLR otherwise (e.g., compare FedSimCLR against Spectral CL without the UV loss)**. Now as to why Spectral CL + UV does better than FedSimCLR, this could be attributed the fact that in the centralized setting, Spectral CL can be better than SimCLR [2] and, through the modifications stemming from our intuitions and theory in the case of MI, can also do better in specific federated settings. Exploring alternative dependency metrics such as [6] in the federated setting is definitely interesting and something we will explore in future work.
>
> -	*Relation to pseudo-labelling approaches*: while from our perspective there are no pseudo-labels, we do agree that having these results as well will help providing a better overall picture. To this end, we are running experiments with two pseudo-label approaches, SemiFed and CBAFed, and so far we observe similar outcomes: the UV loss motivated by our theory is beneficial in the case of label skew.
>
> -	*Supervised baseline*: you are partially correct, in that the supervised baseline corresponds to variant FedAvg. Specifically, we use the superior server-side averaging FedAdam proposed by [1], as we explain in Appendix A. In [1] the authors show empirically that this approach is strictly superior to SCAFFOLD for the cross-device setting, which is why we have not included SCAFFOLD. We have experimented with FedProx but did not see any effect of $\mu>0$ for our choice of $E=1$ across experiments unless setting $\mu>>1$ to detrimental effect. FedBN is a proven FL algorithm under covariate shift, however it prescribes the existence of BatchNorm layers in the network, which we choose to omit. FedBN further assumes the existence of a data-set for computing statistics on a new client, which is an assumption standard FedAvg (FedAdam) does not require with GroupNorm. In summary, we believe that FedAdam is a strong supervised baseline under the given assumptions.
>
> We hope that with the above we have addressed your concerns, and we are happy to continue the discussion otherwise.
>
> [1] Adaptive Federated Optimization, Reddi et al., ICLR 2021
>
> [2] Provable Guarantees for Self-Supervised Deep Learning with Spectral Contrastive Loss, HaoChen et al., NeurIPS 2021

---

> > ### Comment · Reviewer_wQ6s · 2023-11-21
> > **Response to authors**
> >
> > Dear authors,
> >
> > Thank you for your response. Thank you for clarifying that the results for spectral CL are only better with your proposed UV loss, I indeed missed that. With this in mind, it could improve clarity to present the UV loss as the main contribution that can be applied to many methods (it seems to improve SimCLR and spectral CL, but not SimSiam). Independent of the presentation, this sufficiently addresses my concern about the empirical results. Thank you for also including results on pseudo-labeling approaches. Regarding the supervised baseline, using FedAdam is a sensible choice. In my experience, however, it has often been outperformed by SCAFFOLD and FedProx on non-iid data. Therefore, please consider adding one or both methods as baselines in the next version of the manuscript. This is not a critical point, though. The argument for omitting FedBN is sound.
> >
> > Overall, my concerns have been addressed and I will raise my score accordingly.

---

> > > ### Author Response · Authors · 2023-11-22
> > > **Response to reviewer**
> > >
> > > Dear wQ6s,
> > >
> > > We sincerely thank you for the positive feedback and for revising your score. We also uploaded a revision to our submission that includes in the appendix results with the pseudo-labelling baselines where we generally observe that:
> > >
> > > - They tend to underperform compared to the other methods
> > > - They improve by adding the user-verification loss motivated from our mutual information perspective
> > >
> > > Due to the limited amount of time for the rebuttal, we were not able to run any experiments with SCAFFOLD, but we will update the paper appropriately in the next revision. Once again, thank you for helping us improve our work.

---

### Official Review · Reviewer_JEx9 · 2023-11-10

**Soundness:** 3 good
**Presentation:** 3 good
**Contribution:** 2 fair
**Rating:** 6
**Confidence:** 4

**Summary:**

This work extends SimCLR for federated learning, emphasizing multi-view mutual information maximization. It connects contrastive representation learning with user verification and introduces a user verification loss to improve global multi-view mutual information. Additionally, the study extends SimCLR to the federated semi-supervised setting, achieving a supervised SimCLR objective with specific modifications. The research explores the impact of non-i.i.d. data on federated unsupervised learning and shows that the global objective has mixed effects depending on the source of non-i.i.d. data.

**Strengths:**

- The problem of pretraining large models in a federated setting is quite important and has seen little progress so far.
- The proposed LB on the global multi-view objective is principled and as the authors show amenable to federated training.
- Experiments in the semi-supervised setting are a nice addition to the paper, and clearly shows that their objective can be built upon.

**Weaknesses:**

- The paper lacks convergence analysis of their optimization algorithm, which is quite common in FL papers.
- Experiments on more challenging/heterogeneous benchmarks like ImageNet are missing.
- Discussion on how their objective can be adapted to other centralized pretraining objectives is missing. (See questions)
- (Minor/Nit) Proposition 2 need not be stated, it follows immediately from previous Lemmas.

**Questions:**

- How does the proposed MI LB/relaxation work if we move slightly away from SimCLR and look at related objectives: InfoNCE, or even non-contrastive ones like Barlow Twins? Are federated versions of these similar to Federated SimCLR?

---

> ### Author Response · Authors · 2023-11-17
> **Response to review**
>
> Thank you for reviewing our work and for acknowledging the theoretical soundness of our proposed methods as well as the efficacy of our semi-supervised setting. Regarding your specific points:
>
> -	*Complexity of the experiments*: we chose these settings as they are the most popular settings considered in the federated SSL literature. Furthermore, while the difficulty of the image dataset itself is lower compared to settings typically considered in centralized SSL (i.e., ImageNet), they are still difficult from the perspective of federated learning due to the extreme non-i.i.d.-ness we consider, e.g., class proportions from Dirichlet sampling with $\alpha = 0.01$ together with covariate shift, i.e., our joint shift setting.
>
> -	*Applying our objective to other centralized pretraining objectives*: we have indeed demonstrated how our insights apply to other pretraining objectives, be it contrastive, i.e., SimCLR (which is an instance of InfoNCE) and Spectral CL, or non-contrastive, i.e., SimSiam, by intuitively using the same structure for the loss (see the paragraph above the “Unsupervised Setting” in the experiment section). We will expand more about this in the appendix. Indeed, in doing so we observe similar patterns: In the fully unsupervised case with Spectral CL we have better performance in the presence of label skew when adding the UV loss. We also extended these baselines in the same way to the semi-supervised setting, i.e., applying the pretraining objective between elements that belong to the same class while adding an auxiliary classification head that predicts the correct class for those datapoints where it is available. In this scenario we also observe similar improvements, with our SimSiam modifications and the UV loss we significantly improve performance. Extending to feature contrastive approaches such as Barlow Twins is an interesting direction for future work.
>
> -	*Proposition 2*: it indeed follows from the previous arguments, so we are happy to remove it.
>
> We hope that with the shared response and the above, we convince you about the contributions of our work. Let us know if there is something that we should further elaborate upon.

---

> > ### Comment · Reviewer_JEx9 · 2023-11-22
> > **Response to Rebuttal**
> >
> > I thank the reviewers for the additional experiments on SimSiam and Spectral CL. This clarifies the generality of their approach to other SSL objectives. Given this, I feel this paper presents a useful contribution to the field by providing a principled lower bound  that decomposes the SSL objective into a local objective that only uses client data and a user-verification loss.
> >
> > Still, I would encourage the authors to consider adding (in the order of priority): i) experiments on large scale datasets like ImageNet, or medium scale TinyImageNet, since SSL pretraining is often performed at this scale at least, and ii) convergence analysis for the proposed federated optimization objective under assumptions on task diversity, at least in a simplified setting where the SSL objective is convex.
> >
> > For now, I will retain my score and will re-evaluate post discussion with other reviewers. I thank the authors again for their efforts to improve the paper.

---

> ### Author Response · Authors · 2023-11-22
> **Response to reviewer**
>
> We would like to thank you for acknowledging the generality of our approach, the usefulness of our contribution to the federated SSL field and the additional feedback. Based on your suggestions, we uploaded a revision to our manuscript that includes some very preliminary (i.e., not fully converged and with only one random seed) results on unsupervised and semi-supervised training on TinyImagenet in the Appendix. The preliminary observations are:
>
> - In the unsupervised setting, the user-verification loss is generally beneficial for some models in the cases where we expect it to be
> - In the semi-supervised setting, the user-verification loss is still beneficial for the label skew. Adding the rotation non-i.i.d.-ness in this semi-supervised setting resulted in a very difficult problem where with 10% of the labels the convergence is very slow so we were not able to get meaningful performance with any method during the first 15k rounds.
>
> We will continue running these experiments and will update, once we are able, our submission accordingly. Once again, we thank you for the positive feedback and for helping us to further improve our work.

---

> > ### Comment · Reviewer_JEx9 · 2023-11-22
> > **Response to Rebuttal**
> >
> > Thank you for adding the TinyImagenet experiments in such a short time frame. Table 4 in B.1 suggests that Local SimCLR and Federated SimCLR perform similarly. Also, it is unclear if adding UV loss helps since it seems to hurt for all shift types on SimSiam (for both CIFAR-10 and tinyImagenet).
> >
> > I understand that these are initial results, but unfortunately in the current form do not seem to suggest that the proposed objective is helpful for datasets of the scale of TinyImagenet, at least in the unsupervised case. For the semi-supervised setting there are still some gains observed (though still marginal). So, I encourage authors to consider updating the final version with more statistically significant and carefully hyperparameter tuned performance measurements for their own method and the baselines.
> >
> > At the same time, I believe that the reported marginal gains are OK, if the method has other benefits in the federated setting, like faster convergence/fewer communication rounds. Maybe I missed discussion on the latter in the paper, please let me know if that is the case.

---

> > > ### Author Response · Authors · 2023-11-22
> > > **Dear reviewer JEx9**
> > >
> > > We appreciate your fast response to our additional results. Unfortunately, given our limited compute resources, more than preliminary results cannot be done in the timeframe of this rebuttal. We generally observe that the benefit of the UV loss manifests later in training. The federated nature of our simulation makes these experiments take approximately two weeks for 30k rounds and even though we started them as soon as possible, we cannot wait for full convergence in the time for this rebuttal.
> > >
> > > We would kindly ask the reviewer to not dismiss the evidence we bring for the other datasets and - in fact - already show the benefits of our method for TinyImagenet when considering Spectral CL in the unsupervised setting as well as across all methods for the semi-supervised setting, **without any hyper-parameter tuning**. We believe to have addressed SimSiam performance in the main text (section 4), please let us know if there are remaining questions.
> > >
> > > Rest assured that we will update the camera-ready version of the paper with a full suite of numbers to satisfy our own demands for rigorous experimental evidence.
> > >
> > > The benefits of our method can be condensed to allowing the federated training procedure to exploit correlation between the source of non-iid-ness and the downstream task in the semi/un-supervised setting. We make no claims about convergence or communication rounds - in fact, we inherit theoretical guarantees from FedAvg directly.
> > >
> > > Thanks for spending the time on our paper

---

### Author Response · Authors · 2023-11-17
**Shared response to reviewers**

We thank the reviewers for taking the time to review our work and for providing valuable and constructive feedback. We especially appreciate that the reviewers acknowledge the theoretical soundness and presentation of our work, the importance of the task we consider, and our thorough evaluation.

We would first like to state and emphasize the contributions of our work clearly:
We investigate and theoretically analyze federated contrastive learning through the lens of multi-view¬ mutual information (MI), with SimCLR as the running example. Through our analysis, we make several novel contributions, be it via prior works [Proposition 1, Lemma 2.1] or our own [Lemma 2.2, Proposition 3]:

1.	We show that the simple local SimCLR (frequently used as a baseline on federated SSL works [1, 2]) maximizes the client conditional MI. We also analyze the gap to the global MI and show that it can be bounded via a user-verification (UV) loss.

2.	Non-i.i.d.-ness is one of the main challenges in FL. Therefore, we theoretically state the importance of the UV loss in specific non-i.i.d. settings [Proposition 3]; we show that the UV objective is a lower bound to the (unknown) label classification objective in the case of label-skew. While our exposition focused on the perspective of MI and SimCLR, **we show that this result is general and applies to other unsupervised methods**. Therefore, **we believe it is a significant contribution and of independent interest to the federated SSL literature**. Indeed, we empirically see that **there is significant improvement to SimCLR** (up to ~6% for unsupervised and ~4% for semi-supervised), **Spectral CL** (up to ~11% for unsupervised and ~5% for semi-supervised), and **SimSiam** (up for ~5% for semi-supervised) **with the UV loss** in the case of label skew. On the flip side, we also show that **the simple local variants of the unsupervised methods frequently used as baselines in the federated SSL literature can work better in the case of covariate shift** compared to variants that target optimizing a global loss. We believe **this is another important observation for the federated SSL community**, especially given that much of the related work, e.g., [1, 2], focuses on closing the gap between local and global pretraining. We reorganized the results in the tables to highlight the above better.

3.	From the MI perspective, we also extend SimCLR to the case of partially available labels. **This extension is a novel result** that leads to simple modifications to the architecture/objectives, **does not need complex hyperparameter tuning**, unlike pseudo-labeling approaches, and demonstrates a connection of SimCLR to hard negative mining. Once more, **while this result was derived from the perspective of MI and SimCLR, it also applies to other methods**, as we demonstrate empirically. More specifically, **both other contrastive methods**, such as Spectral CL, **and non-contrastive methods**, such as SimSiam, **can use a similar strategy to perform effective semi-supervised learning**. **We believe this is another significant contribution of our work**, which can lead to novel research directions in general for federated semi-supervised learning, as mentioned by reviewer JEx9.
We have updated our exposition to communicate our work's core contributions better.

Finally, one more common point raised by the reviewers is the need for a convergence analysis. Through our MI perspective, the objectives we arrive at are simple averages of local objective functions involving the server model parameters, which we optimize with FedAdam [3]. Therefore, the standard convergence rates from [3] apply, and no additional convergence analysis is required. We have added that conclusion to our paper.

---

> ### Author Response · Authors · 2023-11-17
> **Shared response to reviewers**
>
> Following the reviewers’ suggestions, we have implemented and are currently running the following additional experiments:
>
> -	Pseudo-labelling baselines based on SemiFed [4] and CBAFed [5] for the semi-supervised setup. Baselines such as the ones mentioned by w6Qs are not compatible with our setup, due to assuming the existence of a publicly available dataset. Learning curves suggest that with SemiFed the performance is similar / a bit worse than supervised federated training on just the labelled subset of the data. However, when adding the UV loss to SemiFed as well, we once again see improvements in performance in the case of label skew, surpassing the baseline supervised training. For CBAFed, we adopted the modification with partially labeled clients (see Appendix F.5 of [5]). As we observe, the training speed of CBAFed in cross-device setting is significantly lower than of our semi-supervised Federated SimCLR.
>
> -	FeatARC [6] (without the clustering step, in order to be consistent with our setup) for the purely self-supervised scenario. Learning curves currently suggest that FeatARC (“Align Only”) in our joint-shift cross-client Cifar10 FL setting does not improve upon vanilla local SimCLR. We chose regularization strength $\lambda=0.5$ since the default suggestion of $1$ caused loss divergence.
>
> We have annotated major changes to the first version in blue. Additionally, please find new algorithms in the appendix as well as reorganized Table 1 and 2.
>
> [1] Federated Unsupervised Representation Learning, Zhang et al., 2020
>
> [2] Federated Contrastive Learning for Decentralized Unlabeled Medical Images, 2021
>
> [3] Adaptive Federated Optimization, Reddi et al., ICLR 2021
>
> [4] SemiFed: Semi-supervised Federated Learning with Consistency and Pseudo-Labeling, Lin et al., 2021
>
> [5] Class Balanced Adaptive Pseudo Labeling for Federated Semi-Supervised Learning, Li et al., CVPR 2023
>
> [6] Does Learning from Decentralized Non-IID Unlabeled Data Benefit from Self Supervision, Wang, Lirui, et al. ICLR 2022

---

> > ### Author Response · Authors · 2023-11-22
> > **Shared response to reviewers**
> >
> > Dear reviewers,
> >
> > We would like to let you know that we have uploaded a new revision of our manuscript that includes the results from our additional experiments along with preliminary results on TinyImagenet.

---

### Meta-Review · Area_Chair_knTt · 2023-12-09

**Metareview:**

All reviewers appreciate the work, which extends SimCLR for federated learning, emphasizing multi-view mutual information maximization. The SimCLR extension to FL is principled and solid experiments --both prior to submission and during the (short) response time--demonstrate the efficacy of the approach. It's a solid work and a nice contribution.

**Justification For Why Not Higher Score:**

No convergence analysis.

**Justification For Why Not Lower Score:**

The experiments deserve a spotlight.

---

### Decision · Program_Chairs · 2024-01-16

Accept (spotlight)